# Do Large Language Models Have Compositional Ability? An Investigation into Limitations and Scalability

**Zhuoyan Xu,*** **Zhenmei Shi,*** **Yingyu Liang**
University of Wisconsin–Madison
{zxu444,zhmeishi,yliang}@cs.wisc.edu

## Abstract

Large language models (LLMs) have emerged as powerful tools for many AI problems and exhibit remarkable in-context learning (ICL) capabilities. Compositional ability, solving unseen complex tasks that combine two or more simple tasks, is an essential reasoning ability for Artificial General Intelligence. Despite the tremendous success of LLMs, how they approach composite tasks, especially those not encountered during the pretraining phase, remains an open and largely underexplored question. In this study, we delve into the ICL capabilities of LLMs on composite tasks, with only simple tasks as in-context examples. We develop a test suite of composite tasks including linguistic and logical challenges and perform empirical studies across different LLM families. We observe that models exhibit divergent behaviors: (1) For simpler composite tasks that apply distinct mapping mechanisms to different input segments, the models demonstrate decent compositional ability, while scaling up the model enhances this ability; (2) for more complex composite tasks involving reasoning multiple steps, where each step represents one task, models typically underperform, and scaling up generally provides no improvements. We offer theoretical analysis in a simplified setting, explaining that models exhibit compositional capability when the task handles different input parts separately. We believe our work sheds new light on the capabilities of LLMs in solving composite tasks regarding the nature of the tasks and model scale. Our dataset and code are available at https://github.com/OliverXUZY/LLM_Compose.

## 1 Introduction

In recent years, large language models (LLMs) have revolutionized the natural language processing (NLP) and general AI community. Recent advances, including ChatGPT (OpenAI, 2022), GPT4 (OpenAI, 2023), and Claude 3 (Anthropic, 2024) have shown success in various fields. As model scale increases, larger models exhibit new behavior known as emergence ability. One remarkable emergence is the in-context learning ability (ICL) (Brown et al., 2020), where a model can solve new tasks given only a few examples as input, without any parameter updates. However, despite recent success, how LLMs solve complex reasoning tasks, particularly not seen in pre-training, remains an open question and largely lacks understanding.

In this paper, we focus on the problem of how LLMs tackle composite tasks that incorporate multiple simple tasks. Specifically, we investigate whether a model trained and in-context learned on individual tasks can effectively integrate these skills to tackle combined challenges, which are intuitive and simple for humans. For instance, in Figure 1, if a human is given examples where words following an asterisk (*) will be capitalized and words surrounded by parenthesis will be permuted, one can also understand words following an asterisk (*) surrounded by parenthesis will be capitalized and permuted simultaneously. This basic generalization seems trivial, yet we observe that LLMs fail to generalize in this way.

---

*Equal contribution

Compositional ability is an active problem in the AI community and is crucial for advancing Artificial General Intelligence (AGI). Recent studies have made significant contributions to the understanding of this area. Dziri et al. (2023) formulate compositional tasks as computation graphs to quantify each task's complexity level. Power et al. (2022) show that models may develop generalization capabilities when trained extensively, beyond the point of overfitting, highlighting a phenomenon known as "grokking". An et al. (2023b) examines how LLMs acquire abstract reasoning and achieve compositional generalization in a linguistic context through ICL by testing LLMs on tasks that involve translating a formal language with a custom grammar. Although these studies offer insight, how LLMs compose tasks together is still not fully understood, especially in the ICL setting. Moreover, the absence of a solid theoretical framework in these discussions needs to be investigated concerning the underlying mechanisms of such behaviors.

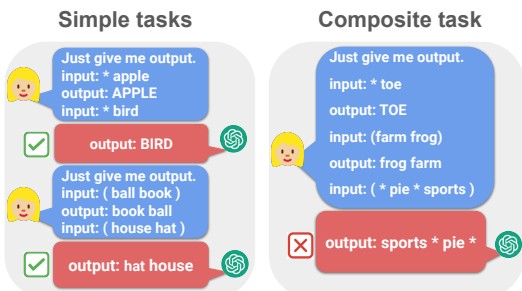

Figure 1: Inconsistent performance in GPT-4. Consider two simple tasks: If a word is followed by an asterisk (*), capitalize the letter. If two words are surrounded by parentheses, swap the positions. GPT-4 correctly solves two simple tasks based on demonstrations (left). The composite tasks have test inputs with both asterisk (*) and parenthesis. The correct answer should be *output: SPORTS PIE*. However, GPT-4 fails to solve the composite tasks (right). The same failure was observed in Claude 3.

Inspired by these seminal works, we further evaluate LLMs on a series of compositional tasks through ICL. The models were presented with examples of simple tasks and then asked to tackle composite tasks that they had not encountered during pretraining or in-context learning. We observe various behaviors: (1) for some composite tasks, the models showed a reasonable level of compositional skill, a capability that improved with larger model sizes; (2) for more complex composite tasks requiring sequential reasoning, the model struggle, and increasing the model size typically did not lead to better performance.

Our key intuition is that if the simple tasks that form a composite task can be easily separated into subtasks based on the inputs (e.g., performed separately on different parts of the input sentence), the model is more likely to complete such a composite task successfully (we call it "a separable composite task"). The performance of the model depends on how it connects and uses the information given for each part of the task. To clarify this insight, we present theoretical analyses in a simplified setting and provide key insights into conditions needed for success in such separable composite tasks.

Our contributions are twofold. **Empirically**, we introduce a variety of composite tasks from both the linguistic and logical domains to explore how the nature of these tasks influences the compositional performance of LLMs through ICL. **Theoretically**, we provide analysis on a simple yet insightful model: a one-layer single-head linear self-attention network (Von Oswald et al., 2023; Akyürek et al., 2023; Mahankali et al., 2023; Zhang et al., 2023b). This framework allows us to demonstrate a clear separation in input embedding, effectively breaking down composite tasks into simpler components. We delve into the scaling of language models by examining the structure of the key and query matrices in the attention mechanism, arguing that larger models with a more complex internal structure exhibit enhanced performance on individual tasks, thereby improving their overall compositional capabilities on such separable tasks.

## 1.1 Related Work

We briefly summarize related work and provide a detailed discussion in Appendix A.

LLMs are often Transformer-based (Vaswani et al., 2017) equipped with the enormous size of parameters and pretrained on vast training data. One key property that makes such LLM successful is called scaling law: Increasing the scale of language models (pretraining data scale, model parameters) can lead to better performance in downstream tasks. One ability that emerges as the model scale increases is in-context learning (ICL) (Brown et al., 2020). Given a sequence of labeled examples and a testing example (combined as a prompt), the model can construct new predictors for testing examples without further parameter updates Dong et al. (2022). Scaling law was first proposed by Kaplan et al. (2020) and then followed up by Hoffmann et al. (2022), emphasizing both the scale of models and training data. Recent works show LLMs with larger scales have distinct behaviors compared to smaller language models (Wei et al., 2023b; Shi et al., 2023b). This work investigates experiments and analyses how LLM can exhibit compositional ability in ICL.

Solving complex tasks and reasoning is an active problem in the AI community Huang & Chang (2022). There is a line of empirical works investigating the compositional ability in linguistic fashion (Kim & Linzen, 2020; Levy et al., 2022; An et al., 2023a;b). LLMs are capable of learning abstract reasoning (e.g., grammar) to perform new tasks when finetuned or given suitable in-context examples. In our work, we include linguistic experiments as part of our testing suite, illustrating LLMs' compositional ability. Ye et al. (2023); Berglund et al. (2023); Dziri et al. (2023) show LLMs will struggle to solve tasks requiring reasoning. Berglund et al. (2023) studies that LLMs trained on "A is B" fail to learn "B is A". In our work, we conduct similar experiments showing LLMs will fail on composite if different steps of logical rules are mixed.

## 2   Warm-up: A Failure Case for Composition

Our goal is to understand the behavior of LLMs on compositional reasoning tasks. As a warm-up, we evaluate the **Capitalization & Swap** tasks (Figure 1) on different models. Recall the tasks: given words of common objects, **\*** represents the operation of capitalizing the letters, **()** represents swapping the positions of the two words. We consider the standard in-context learning setting, which concatenates input-output examples $K = 10$ and one test input as the prompt for LLM. We perform experiments across various LLM families, e.g., Llama families (Touvron et al., 2023) and GPTs (Radford et al., 2019; Black et al., 2021), see model details in Appendix B.

**Evaluation settings.** To make thorough evaluations, we consider four settings: (1) capital: only on the capitalization task; (2) swap: only on swap; (3) composite: in-context examples are from simple tasks while the test input is about the composite task; (4) composite in-context: in-context examples and the test input are all drawn from the composite task. The composite in-context setting reduces the evaluation to another simple task, not requiring the model to compose the simple task

|  | Composite | Composite in-context |
|---|---|---|
| Prompt | input: * apple
output: APPLE
input: ( farm frog )
output: frog farm
input: ( * bell * ford ) | input: ( * good * zebra )
output: ZEBRA GOOD
input: ( * bicycle * add ) |
| Truth | output: FORD BELL | output: ADD BICYCLE |

Table 1: Examples of two settings on composite tasks. Composite: in-context examples are about simple tasks, while the test input is about the composite task. Composite in-context: both in-context examples and the test input are about the composite task.

ability but directly learning from the in-context examples. It serves as the gold standard performance for the composite task. See Table 1 for illustration.

**Results.** In Figure 2, somewhat surprisingly, we observe that LLMs cannot solve the composite task, although they perform well on simple tasks. There is a significant gap between the performance in these settings. Models in Llama families can solve capital and swap with nearly ~90% accuracy but only achieve around 20% or below on the composite

task. We also observe that composite in-context examples will significantly improve the performance: The accuracy of Llama families can go up to match the simple task accuracy. These observations show that *the models fail to compose the knowledge from the simple tasks, although they do have the representation power to solve the composite task* (which can only be exploited when provided composite in-context examples) and *scaling up may not help*.

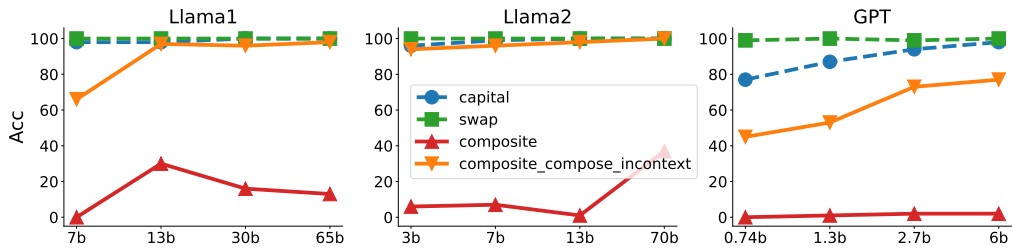

Figure 2: The exact match accuracy ($y$-axis) vs the model scale ($x$-axis, "b" stands for billion) for **Capitalization & Swap** tasks (example in Figure 1). Line *capital*: performance on the simple task of capitalization; *swap*: on the simple task of swap; *composite*: in-context examples are from simple tasks while test input from the composite task. *composite incontext*: in-context examples and test input are all from the composite task (example in Table 1).

## 3 Variability of Compositional Performance

The experiment on Capitalization & Swap shows failure cases while existing studies reported some successful composite abilities (Levy et al., 2022; An et al., 2023b). This observation suggests a more refined perspective: LLMs exhibit variable compositional abilities, excelling in certain composite tasks while struggling in others. This section expands our exploration to additional composite tasks to further examine and understand this variability.

We introduce more composite tasks, including linguistic and logical challenges, wrapped as a testing suite. Similar to the Capitalization & Swap experiment, we design composite tasks that compose two simple tasks and evaluate the model in four settings: the two simple tasks, the composite setting, and the composite in-context setting (Table 1 show examples for the latter two). We consider two kinds of task: logical rules and linguistic translation. We first choose two simple tasks and compose them to construct a composite task.

To address concerns about data leakage and the possibility that models encounter similar tasks during pretraining, we opt for synthetic data in this work. While it is challenging to guarantee that test data has never been seen during pretraining, we take significant steps to mitigate this risk. Specifically, we construct our compositional test data using a unique syntax and mapping mechanism. This approach substantially shifts the data distribution away from existing web-scale data, making it highly improbable that our test data has been encountered during pretraining. By doing so, we aim to create novel composite tasks that comprehensively evaluate the models' compositional abilities.

Section 3.1 investigates logical tasks and Section 3.2 investigates translation tasks.

We perform experiments to answer the following questions: (**Q1**) How do LLMs perform in various tasks, where models might perform well in some scenarios while failing in others? (**Q2**) Does scaling up the model help in general? (**Q3**) Is the variability in performance relevant to the nature of tasks? Our experiments provide the following answers: (**A1**) A pattern of variable performance is observable across a range of composite tasks. (**A2**) Scaling-up helps when the model exhibits compositional ability for certain tasks but may not help when the model initially struggles. (**A3**) In tasks that involve processing inputs from varied segments or perspectives, especially simpler ones, the model tends to demonstrate compositional capabilities.

### 3.1 Composite Logical Rules

We enhance our suite of logical tasks by introducing a series of straightforward tasks that process either simple words or numerical values, with the output being a specific functional transformation of the input. These tasks are detailed in Table 2.

| Tasks | Task | Input | Output |
|-------|------|-------|--------|
| **Words** | (A) Capitalization | apple | APPLE |
| | (B) Swap | bell ford | ford bell |
| | (C) Two Sum | twenty @ eleven | thirty-one |
| | (D) Past Tense | pay | paid |
| | (E) Opposite | Above | Below |
| **Numerical** | (F) Plus One | 435 | 436 |
| | (G) Modular | 15 @ 6 | 3 |
| | (H) Two Sum Plus One | 12 # 5 | 18 |

Table 2: This table contains a collection of simple logical tasks. The *Words* category encompasses tasks that modify words at the character or structural level. The *Numerical* category is devoted to tasks that involve arithmetic computations performed on numbers.

Composite tasks are created by merging two simple tasks. We conceptualize simple tasks as functions, $f(\cdot)$ and $g(\cdot)$ that map inputs to their respective outputs. We identify two distinct approaches to creating composite tasks: **(1) Compose by parts**: For inputs $x, y$, the result is $f(x), g(y)$. One example is **(A) + (F)** in Table 3. If a numerical number is given, it will increment by one; if the word is given, the letters will be capitalized; if both are given, perform both operations.

| Tasks | Simple Task | Simple Task | Composite |
|-------|-------------|-------------|-----------|
| **(A) + (B)** | input: * apple
output: APPLE | input: ( farm frog )
output: frog farm | input: ( * bell * ford )
output: FORD BELL |
| **(A) + (F)** | input: 435
output: 436 | input: cow
output: COW | input: 684 cat
output: 685 CAT |

Table 3: Examples of the two logical composite tasks. Full examples can be found in Appendix B.

**(2) Compose by steps**: Given input $x$, the result is $f(g(x))$. One example is **(A) + (B)** in Table 3. We use customized symbols as function mapping for composing two simple tasks. Examples are in Figure 1 and Table 3. Following existing work, we use exact match accuracy to evaluate the performance since the output for these tasks is usually simple and short.

**Results.** We provide our main results on composite tasks in Table 4. For the composed by parts tasks **(A) + (F)** and **(D) + (F)**, the models show strong compositional ability: the composite accuracy is high, improves with increasing scale, and eventually reaches similar performance as the "gold standard" composite in-context setting, as highlighted in red numbers. We refer to these tasks as "separable composite tasks", which are relatively easy for the model to solve. On the compose-by-step tasks, we observe the models have various performances. For composite tasks with sequential reasoning steps, the models exhibit various performances. For tasks involving capitalization **(A)** or swap **(B)**, the model has poor performance on a small scale (7b or lower) but has increased performance in increased model scale, such as 44% accuracy in **(A) + (C)** and 66% accuracy in **(B) + (D)**. One exception is Llama1-65b, which has lower accuracy than a smaller-scale model. We conjecture it is due to some unknown inductive bias during the pretraining. On composite steps tasks involving the arithmetic calculation of numerical numbers **(G) + (H)**, the model has the worst performance, and increasing the model scale does not provide benefits. A key observation is that compose-by-part tasks are separable compositions where the input can be broken down into two distinct segments. Such tasks are typically straightforward for a model to address. In all experiments, providing composed examples as in-context demonstrations

will help the model understand and solve the composite tasks well, such as **Com. in-context** rows in all task combinations. We conclude that models fail to compose mechanisms of two simple tasks together; however, given composite examples, models can learn the composed mechanism efficiently. We also experimented with prompt demonstrations and found instructions provide no direct results; see more experimental details in Appendix B.2. See more experimental results (including Llama3 (Meta, 2024)) and visualizations in Appendix B.3.

| | | Mistral | | Llama2 | | | Llama1 | | | |
|---|---|---|---|---|---|---|---|---|---|---|
| | Tasks | 7B | 8x7B | 7B | 13B | 70B | 7B | 13B | 30B | 65B |
| **(A) + (B)** | Capitalization | 99 | 98 | 99 | 100 | 100 | 98 | 98 | 100 | 100 |
| | swap | 100 | 100 | 100 | 100 | 100 | 100 | 100 | 100 | 100 |
| | Compose | 16 | 42 | 7 | 1 | 37 | 0 | 30 | 16 | 13 |
| | Com. in-context | 95 | 96 | 96 | 98 | 100 | 66 | 97 | 96 | 98 |
| **(A) + (C)** | twoSum | 71 | 100 | 72 | 93 | 99 | 62 | 56 | 98 | 99 |
| | Capitalization | 98 | 99 | 100 | 95 | 99 | 97 | 98 | 99 | 99 |
| | Compose | 8 | 19 | 3 | 23 | 44 | 3 | 3 | 31 | 2 |
| | Com. in-context | 31 | 65 | 52 | 77 | 100 | 9 | 22 | 93 | 69 |
| **(A) + (F)** | Capitalization | 97 | 99 | 98 | 77 | 99 | 84 | 96 | 99 | 98 |
| | PlusOne | 100 | 99 | 100 | 100 | 100 | 100 | 100 | 100 | 100 |
| | Compose | 92 | 96 | 74 | 69 | 97 | 57 | 60 | 69 | 99 |
| | Com. in-context | 99 | 98 | 99 | 100 | 100 | 99 | 99 | 100 | 100 |
| **(B) + (D)** | Swap | 100 | 100 | 100 | 100 | 100 | 100 | 100 | 100 | 100 |
| | Past Tense | 97 | 99 | 97 | 100 | 99 | 97 | 98 | 100 | 100 |
| | Compose | 6 | 12 | 0 | 1 | 62 | 57 | 34 | 46 | 5 |
| | Com. in-context | 92 | 98 | 86 | 95 | 98 | 86 | 95 | 89 | 94 |
| **(B) + (E)** | Swap | 100 | 100 | 100 | 100 | 100 | 100 | 100 | 100 | 100 |
| | Opposite | 61 | 62 | 58 | 68 | 65 | 51 | 58 | 64 | 63 |
| | Compose | 0 | 0 | 0 | 0 | 0 | 0 | 0 | 0 | 0 |
| | Com. in-context | 35 | 32 | 12 | 37 | 37 | 0 | 9 | 7 | 9 |
| **(D) + (F)** | Past Tense | 100 | 100 | 98 | 100 | 100 | 100 | 100 | 100 | 100 |
| | Plus One | 100 | 100 | 100 | 100 | 100 | 99 | 100 | 100 | 100 |
| | Compose | 71 | 46 | 32 | 80 | 80 | 40 | 44 | 14 | 74 |
| | Com. in-context | 98 | 100 | 98 | 99 | 100 | 95 | 96 | 98 | 100 |
| **(G) + (H)** | Modular | 25 | 22 | 5 | 23 | 43 | 9 | 16 | 29 | 29 |
| | twoSumPlus | 38 | 42 | 3 | 77 | 90 | 14 | 10 | 40 | 87 |
| | Compose | 4 | 5 | 0 | 1 | 1 | 0 | 0 | 0 | 5 |
| | Com. in-context | 4 | 8 | 13 | 13 | 12 | 11 | 13 | 7 | 12 |

Table 4: Results are evaluated composite tasks on various models. The accuracy is in %.

## 3.2 Composite Linguistic Translation

Inspired by previous work in compositional generalization (An et al., 2023b; Levy et al., 2022; An et al., 2023a; Kim & Linzen, 2020), here we design our composite tasks by formal language translation tasks.

Our translation tasks are mainly derived from semantic parsing task COGS (Kim & Linzen, 2020) and compositional generalization task COFE An et al. (2023b). These two datasets contain input as natural English sentences and output as a chain-ruled sentence following a customized grammar (see details in Appendix C). We construct two composite tasks centered on compositional generalization utilizing the training datasets to create in-context examples. See details in Appendix C.

We use the word error rate (WER) as the metric. It measures the minimum number of editing operations (deletion, insertion, and substitution) required to transform one sentence into another and is common for speech recognition or machine translation evaluations.

**(T1) Phrase Recombination with Longer Chain.** COFE proposed two compositional generalization tasks (Figure 2 in An et al. (2023b)). *Phrase Recombination*: integrate a prepositional phrase (e.g., "A in B") into a specific grammatical role (e.g., "subject", "object"); *Longer Chain*: Extend the tail of the logical form in sentences. We see them as simple tasks, and merge them to form a composite task: substitute the sentence subject in the Longer Chain task with a prepositional phrase from the Phrase Recombination task. Details and examples are in Table 9 of Appendix C.

**(T2) Passive to Active and Object to Subject Transformation.** We consider two tasks from Kim & Linzen (2020). *Passive to Active*: Transitioning sentences from passive to active voice. *Object to Subject*: Changing the same object (a common noun) from objective to subjective. They are merged to form our composite task, where both transformations are applied simultaneously to the input sentence. Details and examples are in Table 8 of Appendix C.

**Results.** Figure 3 shows that LLMs can handle these composite tasks. The WER on the composite task is decent and improves with increasing model scale, particularly in Llamma2 models. These confirm the composite abilities of the models in these tasks.

Here, we notice that both composite tasks are separable composite tasks. If we break down these sentences into sub-sentences and phrases, the simple task operations occur in different parts or perspectives of the input sentences. So, the results here provide further support for composite abilities on separable composite tasks, where simple tasks that form the composite task are related to inputs from different parts or perspectives.

We also observed the LLM exhibits better compositional ability on linguistic tasks than on logical tasks. We conclude natural language inputs can indeed help language models understand concepts better than special symbols or code. Natural language provides a richer context, which aligns better with how these models are trained on large text corpora. In contrast, logical and numerical tasks often rely on more rigid structures, which makes it harder for models to generalize without explicit training on such patterns.

**Discussion.** We observe the capability of models to handle composite tasks is significantly influenced by the task characteristics. If composite tasks contain simple tasks related to different parts or perspectives of the input, the model will tackle the composite tasks well.

One natural explanation is that the model processes the input in some hidden embedding space and decomposes the embedding of the

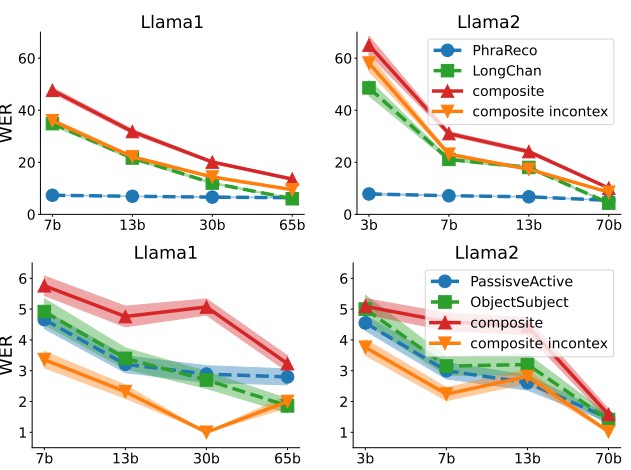

Figure 3: The word error rate (WER) vs the model scale on composite linguistic translation tasks. Dashed lines: simple tasks. Solid lines: composite tasks. Rows: **(T1)** Phrase Recombination with Longer Chain; **(T2)** Passive to Active and Object to Subject Transformation. Columns: different models. Lines: performance in different evaluation settings, e.g., the two simple tasks, the composite setting, and the composite in-context setting (examples are shown in Appendix C).

input into different "regions". Here, each region is dedicated to specific types of information and thus related to different tasks, such as word-level modifications, arithmetic calculations, mapping mechanisms, semantic categorization, linguistic acceptability, or sentiment analysis. Then, suppose the two simple tasks correspond to two different task types that relate to separate regions of the embedding. In that case, the model can effectively manage

the composite task by addressing each simple task operation within its corresponding region. As the model scales, its ability to handle individual tasks improves, leading to enhanced performance on composite tasks in such scenarios. For separable composite tasks, the inputs are divided into distinct regions and also reflected in embeddings, resulting in the model's high performance. However, when the simple tasks are not separable (e.g., requiring sequential steps in reasoning), their information mixes together, complicating the model's ability to discern and process them distinctly. Such overlap often leads to the model's inability to solve the composite task. This intuition is formalized in the following sections in a stylized theoretical setting.

# 4 Theoretical Analysis

## 4.1 Problem Setup

Despite the complex nature of non-linearity in transformers in LLMs, we note it is useful to appeal to the simple case of linear models to see if there are parallel insights that can help us better understand the phenomenon. In this section, we analyze a linear attention module and aim to provide rigorous proof about why LLMs can achieve compositional ability in some simple cases that could shed light on the more intricate behaviors observed in LLMs.

**In-context learning.** We follow existing work (Akyürek et al., 2023; Garg et al., 2022; Mahankali et al., 2023) with slight generalization to $K$ simple tasks. A labeled example is denoted as $(x, y)$ where $x \in \mathbb{R}^d, y \in \mathbb{R}^K$. In a simple task $k \in [K]$, $y$ has only one non-zero entry $y^{(k)}$. In a composite task, $y$ can have non-zero entries in dimensions corresponding to the combined simple tasks. The model takes a prompt $(x_1, y_1, \ldots, x_N, y_N, x_q)$ as input, which contains $N$ in-context examples $(x_i, y_i)$'s and a query $x_q$, and aims to predict $\hat{y}_q$ close to the true label $y_q$ for $x_q$. The prompt is usually stacked into an embedding matrix:
$E := \begin{pmatrix} x_1 & x_2 & \cdots & x_N & x_q \\ y_1 & y_2 & \cdots & y_N & 0 \end{pmatrix} \in \mathbb{R}^{d_e \times (N+1)}$ where $d_e = d + K$. In in-context learning, we first pretrain the model using training prompts and then evaluate the model with evaluation prompts; see details below.

**Pretraining procedure.** We have $B$ training data indexed by $\tau$, each containing an input prompt $(x_{\tau,1}, y_{\tau,1}, \ldots, x_{\tau,N}, y_{\tau,N}, x_{\tau,q})$ and a corresponding true label $y_{\tau,q}$. Consider the following empirical loss: $\hat{L}(\theta) = \sum_{k=1}^{K} \hat{L}_k(\theta) = \frac{1}{2B} \sum_{\tau=1}^{B} \|\hat{y}_{\tau,q} - y_{\tau,q}\|^2$ and the population loss (i.e., $B \to \infty$): $L(\theta) = \frac{1}{2} \mathbb{E}_{x_{\tau,1}, y_{\tau,1}, \cdots, x_{\tau,N}, y_{\tau,N}, x_{\tau,q}} \left[ \left( \hat{y}_{\tau,q} - y_{\tau,q} \right)^2 \right]$.

**Evaluation procedure.** We now detail how to evaluate the model on downstream *composite* tasks. We consider the downstream classification task to be a multi-class classification problem, where the output label is a $K$-dimensional vector, and each entry corresponds to a simple binary classification task. For any given simple task $k$, the binary classification label is given by $\text{sgn}(y_q^{(k)})$, where sgn is the sign function. Similarly, our prediction is $\hat{y}_q^{(k)} = \text{sgn}\left( \hat{y}_q^{(k)} \right)$. The accuracy of a composite task is defined as $\text{Acc}_\theta(x_1, \ldots, y_N, x_q) = \frac{1}{K} \sum_{k=1}^{K} \mathbb{1} \left( \text{sgn}\left( \hat{y}_q^{(k)} \right) = \text{sgn}(y_q^{(k)}) \right)$. We denote it as $\text{Acc}_\theta(\{x_i, y_i\}_{i=1}^{N})$. Here we denote the model performance on each task as separate dimension, (e.g.letter capitalization, numbers increment), and the performance of composite tasks as the aggregation of multiple dimensions.

**Data.** Assume $x \overset{\text{i.i.d.}}{\sim} \mathcal{N}(0, \Lambda)$, where $\Lambda \in \mathbb{R}^{d \times d}$ is the covariance matrix. Assume $y = Wx$, where $W \in \mathbb{R}^{K \times d}$. For any simple task $k \in [K]$, its label is the $k$-th entry of $y$, which is $y^{(k)} = \langle w^{(k)}, x \rangle$, where $w^{(k)}$ is the $k$-th row of $W$. We assume each task weight $w^{(k)} \overset{\text{i.i.d.}}{\sim} \mathcal{N}(0, I_d)$.

**Linear self-attention networks.** These networks are widely studied (Von Oswald et al., 2023; Akyürek et al., 2023; Garg et al., 2022; Zhang et al., 2023b; Shi et al., 2023b). Following them, we consider the following linear self-attention network with parameters

$\theta = (W^{PV}, W^{KQ})$: $f_{\text{LSA},\theta}(E) = E + W^{PV}E \cdot \frac{E^\top W^{KQ}E}{N}$. The prediction of the model for $x_q$ is $\hat{y}_q = [f_{\text{LSA},\theta}(E)]_{(d+1):(d+K),N+1}$, the bottom rightmost sub-vector of $f_{\text{LSA},\theta}(E)$ with length $K$.

**Compositional ability.** We now provide a formal definition about *compositional ability* of an LLM on composite tasks.

**Definition 1** (Compositional Ability). *Consider a composite task $\mathcal{T}$ that combines two simple tasks $k$ and $g$. Let $\mathcal{S}_k$ denote $N$ labeled examples from task $k$, and similarly for $\mathcal{S}_g$. Given an $x_q$ from composite task $\mathcal{T}$, we say that the model has compositional ability on $\mathcal{T}$ if the model has higher accuracy using in-context examples from $\mathcal{S}_k \cup \mathcal{S}_g$ than from either single one, i.e.* $\max\{Acc_\theta(\mathcal{S}_k), Acc_\theta(\mathcal{S}_g)\} \leq Acc_\theta(\mathcal{S}_k \cup \mathcal{S}_g)$.

## 4.2 Theoretical Results

In this section, we present our theoretical results. We explain the observation in empirical results through the lens of confined supports in input embeddings corresponding to separate subspaces (modeling separable composition). We provide theoretical justification showing that separable composite task composite tasks whose inputs are composed by components adhere to certain conditions where models exhibit satisfactory performance. Models will fail when such conditions are violated. We first introduce the basic setup and definitions.

**Disjoint subspaces of simple tasks.** Recall that $x$ lies in a $d$-dimensional space where each dimension represents a different characteristic. A simple task may depend only on a subset of these dimensions since its label only depends on a few features. Let $\mathbb{S} = [d]$ represent the dimensions of $x$. For a task $k$, the output $y^{(k)} = \langle w_k, x \rangle$ depends on a subset of dimensions in $x$. Denote this subset by $\mathbb{K} \subseteq \mathbb{S}$ and call it the active index set for task $k$.

In the following, we always assume that the $K$ tasks have disjoint subspaces: for any two tasks $k \neq g$, their active index sets $\mathbb{K}$, and $\mathbb{G}$ are disjoint, i.e., $\mathbb{K} \cap \mathbb{G} = \varnothing$. In practice, the dimensions within $\mathbb{K}$ could be associated with numerical arithmetic operations, while those in $\mathbb{G}$ might pertain to semantic analysis. This illustrates the model's approach to address these tasks in their respective subspaces.

We now introduce a mild assumption regarding the distribution of input embeddings.

**Assumption 1.** *Given two disjoint subspaces $\mathbb{K}$ and $\mathbb{G}$, the covariance matrix $\Lambda$ of the input distribution can be segmented into block matrices $\Lambda_{\mathbb{K}\mathbb{K}}, \Lambda_{\mathbb{K}\mathbb{G}}, \Lambda_{\mathbb{G}\mathbb{K}}$, and $\Lambda_{\mathbb{G}\mathbb{G}}$, then we assume $\sigma_{max}(\Lambda_{\mathbb{K}\mathbb{G}}) = \sigma_{max}(\Lambda_{\mathbb{G}\mathbb{K}}) \leq \epsilon$ for constant $\epsilon$, where $\sigma(\cdot)$ denote the singular value of matrix.*

Assumption 1 implies that for two separate simple tasks, each associated with its respective feature subspace $\mathbb{K}$ and $\mathbb{G}$, the covariance between these two sets of features is bounded by a constant value. This is a natural assumption when inputs of composite tasks can be decomposed into parts. Suppose we have input embeddings from two tasks: arithmetic computations and semantic analysis. This assumption suggests that the feature subspaces of the input embeddings for two tasks are almost independent.

We now define confined support, which means that each task's input embedding only has support within its feature subspace.

**Definition 2** (Confined Support). *We say a task has confined support if the input $x$ only has larger singular values within its active index set. The norm of entries outside the active index set is bounded by a small constant $\delta$.*

This definition shows that each simple task only has large values within its corresponding subsets of dimensions of input embeddings. For example, let $\mathbb{K}$ represent the first $d_1$ dimensions of an input vector $x$, and $\mathbb{G}$ account for the remaining $d_2$ dimensions, with the total dimension being $d = d_1 + d_2$. The examples from task $k$ will have input as $x = (x_1, x_{\delta_1})$ where $x_1 \in \mathbb{R}^{d_1}, x_{\delta_1} \in \mathbb{R}^{d_2}, \|x_{\delta_1}\| \leq \delta$. Similarly, the examples from task $g$ will have inputs as $x = (x_{\delta_2}, x_2)$.

We now present our results of the compositional ability under a confined support of $x$.

**Theorem 1.** *Consider distinct tasks $k$ and $g$ with corresponding examples $\mathcal{S}_k, \mathcal{S}_g$. If two tasks have confined support, and Assumption 1 is true, then with high probability, the model has the*

*compositional ability as defined in Definition 1. Moreover,*

$$Acc_\theta(\mathcal{S}_k) + Acc_\theta(\mathcal{S}_g) \leq Acc_\theta(\mathcal{S}_{k \cup g}).$$

Theorem 1 shows the compositional ability of LLMs to handle composite tasks that integrate two simple tasks, which have confined support in their own feature subspace.

An illustrative case involves the tasks of Capitalization **(A)** & Plus One **(F)** and Past Tense **(D)** & Plus One **(F)**, as depicted in Table 4. These two simple tasks involve word-level modification and arithmetic operation on separate parts of the input. Due to this separation, each task correlates with a specific segment of the input embedding. Therefore, it is observed that these tasks possess confined supports.

We further provide theory illustrating the **necessity of the confined supports**, we demonstrate that when the confined support is violated, simple tasks begin to show variations (indicated by large singular values) across the entire feature subspace of the input embedding. For instance, the composite task of Capitalization (A) & Swap (B), which involves mixed steps in reasoning as shown in Figure 2, shows poor performance of LLMs given both simple tasks' examples as in-context demonstrations. Another example is Modular (G) & Two Sum Plus (H) as shown in the last row of Table 4, where both simple tasks involve multisteps arithmetic operation. These two tasks share the same embedding space support, mixing their variations and causing the model to be unable to effectively address the composite tasks that integrate them. We further substantiate this observation with Corollary 1, which establishes that when two tasks share overlapping support in the embedding space, a scenario can arise where the model fails to demonstrate compositional ability.

**Corollary 1.** *If two tasks do not have confined support, there exists one setting in which we have*

$$Acc_\theta(\mathcal{S}_k) = Acc_\theta(\mathcal{S}_g) = Acc_\theta(\mathcal{S}_{k \cup g}).$$

Corollary 1 demonstrates that a model's failure to solve tasks with mixed steps reasoning, which contains overlapping input embedding spaces, thereby diminishing the model's ability to solve them when presented together.

We also show the **scaling effect**: if simple tasks have confined support, the compositional ability of language models will increase as the model scale increases in Theorem 2 in Appendix D.1. We demonstrate this by showing that the model's accuracy on each simple task improves with a larger model scale. We finally provide a **case study** on confined support for illustration in Appendix D.2. We defer the full proof in Appendix E.

## 5   Conclusion

In this work, we presented a distinct pattern in LLMs' behaviors when tackling composite tasks. We observed that if the composite task can be separated into two simple tasks whose inputs are from distinct perspectives, the models exhibit decent compositional ability. Otherwise, LLMs will struggle, and scaling up the model size may not offer improvement. We illustrated this behavior across a variety of logical and linguistic challenges. We extended our discussion to the role of input embeddings in affecting model performance, providing a theoretical backup that connects the nature of tasks to how inputs are processed. We anticipate that our research will shed light on the compositional capabilities and reasoning of LLMs, and stimulate further exploration in this direction.

## Impact Statements

Our work aims to improve the understanding of in-context learning capabilities of Large Language Models (LLMs) in the handling of composite tasks. Our paper is mostly theoretical in nature, and thus, we foresee no immediate negative ethical impact. We illustrate the empirical behavior of LLMs on complex reasoning tasks and provide a theoretical explanation for it. In the long term, we hope our work may inspire effective algorithm design and better understanding and employment of LLMs.

## Acknowledgements

The work is partially supported by Air Force Grant FA9550-18-1-0166, the National Science Foundation (NSF) Grants 2008559-IIS, CCF-2046710, and 2023239-DMS.

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

# Appendix

## Contents

In this appendix, we provide more related work in Appendix A. We provide more empirical settings and results for logical tasks in Appendix B and linguistic translation tasks in Appendix C. We provide a theory for confined support and model scalability, along with a case study of a toy model in Appendix D. We provide full proof in Appendix E.

# A   Related Work

**Large language model.** LLMs are often Transformer-based (Vaswani et al., 2017) equipped with the enormous size of parameters and pretrained on vast training data. Typical LLMs includes BERT Devlin et al. (2019), PaLM Chowdhery et al. (2022), LLaMA Touvron et al. (2023), ChatGPT (OpenAI, 2022), GPT4 (OpenAI, 2023). Pretraining methods include masked language modeling (Devlin et al., 2019; Liu et al., 2019), contrastive learning (Gao et al., 2021; Shi et al., 2023a; Sun et al., 2023; 2024) and auto-regressive pretraining (Radford et al., 2018; 2019). Several works (Madasu & Srivastava, 2022; Alajrami et al., 2023) investigate the effects of pretraining on language models. Adapting LLMs to various downstream tasks has received significant attention, e.g., adaptor (Hu et al., 2022; 2023; Zhang et al., 2023a; Luo et al., 2024), prompt tuning (Lester et al., 2021; Li & Liang, 2021; Wei et al., 2023a; Gu et al., 2024c), multitask finetuning (Sanh et al., 2022; Wang et al., 2023b; Xu et al., 2023; 2024c), instruction tuning Chung et al. (2022); Mishra et al. (2022), in-context learning (Min et al., 2022b; Dong et al., 2022; Yao et al., 2023), low-rank adaptation Hu et al. (2022); Zeng & Lee (2024); Hu et al. (2024), reinforcement learning from human feedback (RLHF) (Ouyang et al., 2022) and inference acceleration (Gu et al., 2024b;d; Xu et al., 2024a).

**In-context learning.** LLM exhibits a remarkable ability for in-context learning (ICL) (Brown et al., 2020), particularly for generative models. Given a sequence of labeled examples and a testing example (combined as a prompt), the model can construct new predictors for testing examples without further parameter updates. Several empirical studies investigate the behavior of ICLs. Zhao et al. (2021); Holtzman et al. (2021); Lu et al. (2022) formulate the problems and report the sensitivity. Rubin et al. (2022); Liu et al. (2022); Hongjin et al. (2023); Wang et al. (2023a) provide methods to better choose in-context learning examples. Chen et al. (2022); Min et al. (2022a) use meta training with an explicit in-context learning object to boost performance. Theoretically, Xie et al. (2022); Garg et al. (2022) provide a framework to explain the working mechanism of in-context learning. Von Oswald et al. (2023); Akyürek et al. (2023); Mahankali et al. (2023); Zhang et al. (2023b), investigating with linear models, show how transformers can represent gradient descent and conduct linear regression. Based on these works, we provide an analysis showing how LLM can exhibit compositional ability in ICL.

**Emergence of compositional ability.** Scaling law was first proposed by Kaplan et al. (2020) and then followed up by Hoffmann et al. (2022), emphasizing both the scale of models and training data. Sometimes, increasing scale can lead to new behaviors of LLMs, termed *emergent abilities* (Wei et al., 2022; Arora & Goyal, 2023), such as domain generalization Shi et al. (2024a), math reasoning Gu et al. (2024a), spatial reasoning Wang et al. (2024) and so on. Recent works show LLMs with larger scales have distinct behavior compared to smaller language models (Wei et al., 2023b; Shi et al., 2023b; 2024b). These behaviors can have positive or negative effects on performance. Solving complex tasks and reasoning is an active problem in the AI community Huang & Chang (2022). There is a line of empirical works investigating the compositional ability in linguistic fashion (Kim & Linzen, 2020; Levy et al., 2022; An et al., 2023a;b; Xu et al., 2024b). LLMs are capable of learning abstract reasoning (e.g., grammar) to perform new tasks when finetuned or given suitable in-context examples. In our work, we include linguistic experiments as part of our testing suite, illustrating LLMs' compositional ability. Ye et al. (2023); Berglund et al. (2023); Dziri et al. (2023) show LLMs will have difficulties solving tasks that require reasoning. Berglund et al. (2023) studies that LLMs trained on "A is B" fail to learn "B is A". In our work, we conduct similar experiments showing LLMs will fail on composite if different steps of logical rules are mixed.

# B Logical Tasks

## B.1 Task Setup

We provide a comprehensive explanation of logical composite tasks below. Examples can be seen in Table 5.

- **(A) + (B) Capitalization & Swap**, as in Section 2.
- **(A) + (C) Capitalization & Two Sum.** Given words of numerical numbers, **\*** represents the operation of capitalizing, **@** represents summing the two numbers.
- **(G) + (H) Modular & Two Sum Plus.** Given numerical numbers, **@** represents the operation of taking modular, **#** represents to sum the two numbers and then plus one.
- **(A) + (F) Capitalization & Plus One.** If numerical numbers are given, plus one; if words are given, capitalize the word; if both are given, perform both operations.

Among these, **(A) + (F)** performs the two operations on separable parts of the test inputs (i.e., separable composite task).

| Tasks | Simple Task | Simple Task | Composite |
|---|---|---|---|
| **(A) + (B)** | input: * apple
output: APPLE | input: ( farm frog )
output: frog farm | input: ( * bell * ford )
output: FORD BELL |
| **(A) + (C)** | input: * ( five )
output: FIVE | input: twenty @ eleven
output: thirty-one | input: * ( thirty-seven @ sixteen )
output: FIFTY-THREE |
| **(G) + (H)** | input: 15 @ 6
output: 3 | input: 12 # 5
output: 18 | input: 8 # 9 @ 7
Ouput: 4 |
| **(A) + (F)** | input: 435
output: 436 | input: cow
output: COW | input: 684 cat
output: 685 CAT |

Table 5: Examples of the four logical composite tasks. Note that in **(G) + (H)**, the output of the composite task can be either 4 or 11 depending on the order of operations, and we denote both as correct.

We design our logical tasks following the idea of math reasoning and logical rules. The details are shown in Table 5. Our numerical numbers in Table 2 are uniformly randomly chosen from 1 to 1000. The words of numbers in task **(C)** are uniformly randomly chosen from one to one hundred. The words representing objects in Table 2 are uniformly randomly chosen from class names of ImageNet after dividing the phrase (if any) into words. We randomly chose 100 examples in composite testing data in our experiments and replicated the experiments in each setting three times. We fixed the number of in-context examples as $K = 10$ as demonstrations.

## B.2 Experimental Setup

We use exact match accuracy to evaluate the performance between sequence outputs. The calculation of exact match accuracy divided the number of matched words by the length of ground truth.

For Llama models, we use official Llama1 and Llama2 models from Meta (Touvron et al., 2023), we use open_llama_3b_v2 from open OpenLlama (Geng & Liu, 2023). For GPT models, we use GPT2-large from OpenAI (Radford et al., 2019), and we use GPT-neo models for GPT models in other scales from EleutherAI (Black et al., 2021).

We've experimented with prompt demonstrations. Instructional prompts do help ChatGPT and Claude3 (although we haven't quantified the accuracy in large-scale experiments), but they offer limited benefits for current open-source models. On the other hand, we did not have prompt tuning or any other parameter updates during our evaluation.

In our experiments, we provided the model with instructions. Here are instructions of Figure 1, which were prepended to ICL examples. We refer to our codebase for full instructions and results.

> \* is a function before words for swapping the position of 2 words, # is another function after words for capitalizing letters of words.

### B.3  Results

We show a visualization of some logical task accuracy along the increasing to model scale, complement to Table 4.

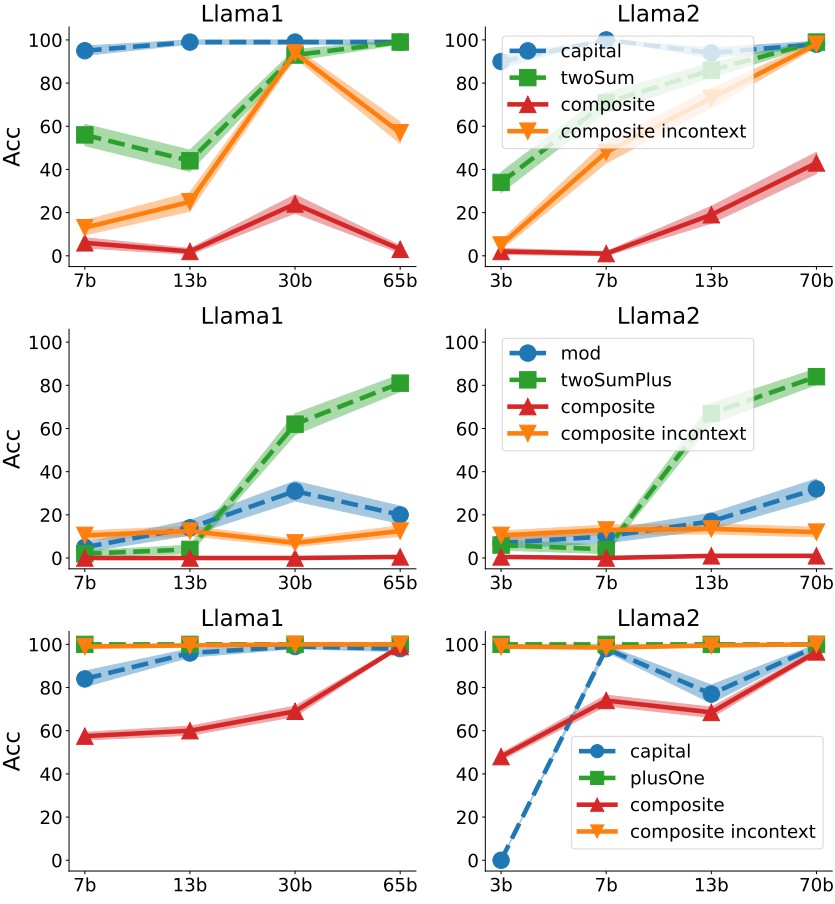

Figure 4: The accuracy v.s. model scale on composite logical rule tasks. Dashed lines: simple tasks. Solid lines: composite tasks. Rows: **(A) + (C)** Capitalization & Two Sum; **(G) + (H)** Modular & Two Sum Plus; **(A) + (F)** Capitalization & Plus One. Columns: different models. Lines: performance in different evaluation settings, i.e., the two simple tasks, the composite setting, and the composite in-context setting (examples for the last two are shown in Table 1).

We also include results for the more recent model Llama3 (Meta, 2024) on the part of our logical tasks to demonstrate the idea. We show results in Table 6.

As shown in the Table 6, for the *separable composite tasks* which are relatively easy for model to solve **(A) + (F)**, the models show strong compositional ability: the composite accuracy is high, improves with increasing scale, and eventually reaches similar performance as the *gold standard* composite in-context setting. For composite tasks with sequential reasoning steps

| | Tasks | Llama3 8B | 70B |
|---|---|---|---|
| **(A) + (B)** | Capitalization | 100 | 100 |
| | swap | 100 | 100 |
| | Compose | 52 | 72 |
| | Com. in-context | 97 | 100 |
| **(A) + (F)** | Capitalization | 100 | 100 |
| | PlusOne | 100 | 100 |
| | Compose | 88 | 100 |
| | Com. in-context | 100 | 100 |

Table 6: Results evaluating composite tasks on Llama3. The accuracy is shown in %.

**(A) + (B)**, the model has poor performance on a small scale but has increased performance on an increased model scale. Providing composed examples as in-context demonstrations will help the model understand and solve the composite tasks well.

## C  Formal Language Translation Tasks

Our translation tasks mainly follow the compositional generalization tasks in COFE (An et al., 2023b). The details can be found in Section 4 in An et al. (2023a). We directly take the source grammar $\mathcal{G}_s$ in COGS, which mimics the English natural language grammar, and reconstruct the target grammar $\mathcal{G}_t$ in COGS to be chain-structured.

We follow the Primitive coverage principle proposed by An et al. (2023b) that primitives contained in each test sample should be fully covered by in-context examples. Here, primitives refer to the basic, indivisible elements of expressions, including subjects, objects, and verbs. Note that multiple sets of in-context examples can meet these criteria for each test case. Across all experimental conditions, we maintain a consistent number of test instances at 800.

We use the word error rate (WER) as the metric. It measures the differences between 2 sentences. It measures the minimum number of editing operations (deletion, insertion, and substitution) required to transform one sentence into another and is common for speech recognition or machine translation evaluations. The computation of WER is divided by the number of operations by the length of ground truth.

| Original Target Grammar | Chain-Structured Target Grammar |
|---|---|
| rose ( x_1 ) AND help . theme ( x_3 , x_1 ) AND help . agent ( x_3 , x_6 ) AND dog ( x_6 ) | HELP ( DOG, ROSE, NONE ) |
| * captain ( x_1 ) ; eat . agent ( x_2 , x_1 ) | EAT ( CAPTION, NONE, NONE ) |
| * dog ( x_4 ) ; hope . agent ( x_1 , Liam ) AND hope . ccomp ( x_1 , x_5 ) AND prefer . agent ( x_5 , x_4 ) | HOPE ( LIAM, NONE, NONE ) CCOMP PREFER ( DOG, NONE, NONE ) |

Table 7: Demonstration in An et al. (2023a) showing examples with the original grammar and the new chain-structured grammar.

In formal language tasks, as mentioned in Section 3.2, we change the original target grammar of COGS to be chain-structured. In Table 7, we list some examples with the original target grammar and the new chain-structured grammar.

- First, to distinguish the input and output tokens, we capitalize all output tokens (e.g., from "rose" to "ROSE").
- Second, we replace the variables (e.g., "x_1") in the original grammar with their corresponding terminals (e.g., "ROSE").
- Then, we group the terminals of AGENT (e.g., "DOG"), THEME (e.g., "ROSE"), and RECIPIENT with their corresponding terminal of PREDICATE (e.g., "HELP") and

combine this group of terminals in a function format, i.e., "PREDICATE ( AGENT, THEME, RECIPIENT )". If the predicate is not equipped with an agent, theme, or recipient in the original grammar, the corresponding new non-terminals (i.e., AGENT, THEME, and RECIPIENT, respectively) in the function format above will be filled with the terminal NONE (e.g., "HELP ( DOG, ROSE, NONE )"). Such a function format is the minimum unit of a CLAUSE.

- Finally, each CLAUSE is concatenated with another CLAUSE by the terminal CCOMP (e.g., "HOPE ( LIAM, NONE, NONE ) CCOMP PREFER ( DOG, NONE, NONE )").

| Task | In-context Example | Testing Example |
|---|---|---|
| Passive to Active | The book was squeezed . 
 SQUEEZE ( NONE , BOOK , NONE ) | Sophia squeezed the donut . 
 SQUEEZE ( SOPHIA , DONUT , NONE ) |
| Object to Subject | Henry liked a cockroach in a box . 
 LIKE ( HENRY , IN ( COCKROACH , BOX ) | A cockroach inflated a boy . 
 INFLATE ( COCKROACH , BOY , NONE ) |
| Composite Task | The book was squeezed . 
 SQUEEZE ( NONE , BOOK , NONE ) 
 Henry liked a cockroach in a box . 
 LIKE ( HENRY , IN ( COCKROACH , BOX ) | A cockroach squeezed the hedgehog . 

 SQUEEZE ( COCKROACH , hedgehog , NONE ) |

Table 8: Testing examples of Passive to Active and Object to Subject, red text shows the verbs changing from passive to active voice in simple tasks, and blue text shows the nouns from objective to subjective.

| Task | | Example |
|---|---|---|
| Phrase Recombination | Input 
 Output | The baby on a tray in the house screamed . 
 SCREAM ( ON ( BABY , IN ( TRAY , HOUSE ) ) , NONE , NONE ) |
| Longer Chain | Input | A girl valued that Samuel admired that a monkey liked that Luna liked that Oliver respected that Savannah hoped that a penguin noticed that Emma noticed that the lawyer noticed that a cake grew . |
| | Output | VALUE ( GIRL , NONE , NONE ) \ 
 CCOMP ADMIRE ( SAMUEL , NONE , NONE ) \ 
 CCOMP LIKE ( MONKEY , NONE , NONE ) \ 
 CCOMP LIKE ( LUNA , NONE , NONE ) \ 
 CCOMP RESPECT ( OLIVER , NONE , NONE ) \ 
 CCOMP HOPE ( SAVANNAH , NONE , NONE ) \ 
 CCOMP NOTICE ( PENGUIN , NONE , NONE ) \ 
 CCOMP NOTICE ( EMMA , NONE , NONE ) \ 
 CCOMP NOTICE ( LAWYER , NONE , NONE ) \ 
 CCOMP GROW ( NONE , CAKE , NONE ) |
| Composite Task | Input | The baby on a tray in the house valued that Samuel admired that a monkey liked that Luna liked that Oliver respected that Savannah hoped that a penguin noticed that Emma noticed that the lawyer noticed that a cake grew . |
| | Output | VALUE ( ON ( BABY , IN ( TRAY , HOUSE ) , NONE , NONE ) \ 
 CCOMP ADMIRE ( SAMUEL , NONE , NONE ) \ 
 CCOMP LIKE ( MONKEY , NONE , NONE ) \ 
 CCOMP LIKE ( LUNA , NONE , NONE ) \ 
 CCOMP RESPECT ( OLIVER , NONE , NONE ) \ 
 CCOMP HOPE ( SAVANNAH , NONE , NONE ) \ 
 CCOMP NOTICE ( PENGUIN , NONE , NONE ) \ 
 CCOMP NOTICE ( EMMA , NONE , NONE ) \ 
 CCOMP NOTICE ( LAWYER , NONE , NONE ) \ 
 CCOMP GROW ( NONE , CAKE , NONE ) |

Table 9: Testing examples of Phrase Recombination and Longer Chain, red text shows the phrase serving as primitives in sentences in simple tasks, and blue text shows the logical structures as sub-sentences in long sentences.

In the following, we provide a detailed explanation of our two composite tasks in translation tasks.

**Passive to Active and Object to Subject Transformation.** Based on the generalization tasks identified in Kim & Linzen (2020)), we select two distinct challenges for our study as two simple tasks. *Passive to Active*: Transitioning sentences from Passive to Active voice. *Object to Subject*: Changing the focus from Object to Subject using common nouns. These

tasks serve as the basis for our composite task, where both transformations are applied simultaneously to the same sentence. Examples illustrating this dual transformation can be found in Table 8.

**Enhanced Phrase Subject with Longer Chain.** COFE proposed two compositional generalization tasks (Figure 2 in An et al. (2023b)): *Phrase Recombination (PhraReco)*: integrate a prepositional phrase (e.g., "A in B") into a specific grammatical role (e.g., "subject", "object"); *Longer Chain (LongChain)*: Extend the tail of the logical form in sentences. We consider these two generalization tasks as two simple tasks, merging them to form a composite task. In particular, we substitute the sentence subject in the Longer Chain task with a prepositional phrase from the Phrase Recombination task, creating a more complex task structure. Detailed examples of this combined task can be found in Table 9.

# D Theory for Confined Support

## D.1 Compositional Ability with Model Scale

We then show that if simple tasks have confined support, the compositional ability of language models will increase as the model scale increases. We demonstrate this by showing that the accuracy of the model on each simple task improves with a larger model scale.

Note that the optimal solutions of the parameter matrices are $W^{*PV}$ and $W^{*KQ}$. We naturally consider that the rank of the parameter matrices $W^{*PV}$ and $W^{*KQ}$ can be seen as a measure of the model's scale. A higher rank in these matrices implies that the model can process and store more information, thereby enhancing its capability. We state the following theorem.

**Theorem 2.** *Suppose a composite task satisfies confined support. Suppose that we have $(x_1, y_1, \ldots, x_N, y_N, x_q)$ as a testing input prompt and the corresponding $W$ where $y_i = Wx_i$. As rank $r$ decreases, $\mathbb{E}_{W, x_1, \cdots, x_N} [Acc_\theta]$ will have a smaller upper bound.*

Theorem 2 shows the expected accuracy of a model on composite tasks is subjected to a lower upper bound as the scale of the model diminishes. This conclusion explains why scaling up helps the performance when the model exhibits compositional ability for certain tasks (those we call "separable composite tasks"). One common characteristic of these tasks is their inputs display confined supports within the embeddings. This is evidenced by the model's decent performance on tasks as presented in Table 4 and Figure 3, where inputs are composed of parts.

## D.2 Case Study of Confined Support

Our theoretical conclusion shows the model behavior regarding input embedding. It states that the model will have compositional ability if tasks are under confined support of input embedding. To illustrate such theoretical concepts and connect them to empirical observations, we specialize the general conclusion to settings that allow easy interpretation of disjoint. In this section, we provide a toy linear case study on classification tasks, showing how confined support on embedding can be decomposed and composite tasks can be solved. We assume $\delta = \epsilon = 0$ in the following simple example.

Consider that there are only two simple tasks for some random objects with the color red and blue and the shape square and round: (1) binary classification based on the color red and blue. (2) binary classification based on shape: circle and square. However, during evaluation, the composite task is a four-class classification, including red circle, red square, blue circle, and blue square.

Then we have two simple tasks $K = 2$. Consider the input embedding $x = (a, b)$, where $a \in \mathbb{R}^2, b \in \mathbb{R}^2, d = 4$. Consider $W = \begin{pmatrix} 1 & -1 & 0 & 0 \\ 0 & 0 & 1 & -1 \end{pmatrix}$ and $y = Wx$.

Consider the inputs from simple and composite tasks as:

- Task 1: Red: $x_1 = (1,0,0,0), y_1 = (1,0)$ and blue: $x_2 = (0,1,0,0), y_2 = (-1,0)$.
- Task 2: Circle $x_3 = (0,0,1,0)$ $y_3 = (0,1)$ and square $x_4 = (0,0,0,1)$ $y_4 = (0,-1)$.
- Composed task: red circle $x_5 = (1,0,1,0), y_5 = (1,1)$, red square $x_6 = (1,0,0,1), y_6 = (1,-1)$, blue circle $x_7 = (0,1,1,0)$ $y_7 = (-1,1)$ and blue square $x_8 = (0,1,0,1)$ $y_8 = (-1,-1)$.

Suppose that we have the optimal solution $\hat{y}_q$ as in Equation (1). Given $x_q = (1,0,1,0)$ as a testing input for a red circle example, During the test, we have different predictions given different in-context examples:

1. Given only examples from Task 1 (red and blue): $[(x_1, y_1), (x_2, y_2)]$, we have $\hat{y}_q = (1,0)$ can only classify the color as red.
2. Given only examples from Task 2 (square and circle): $[(x_4, y_4), (x_3, y_3)]$, we have $\hat{y}_q = (0,1)$ only classify the shape as a circle.
3. Given a mixture of examples from Task 1 and 2 (red and circle): $[(x_1, y_1), (x_3, y_3)]$, we have $\hat{y}_q = (1,1)$ can classify as red and circle.

We can see that in the final setting, the model shows compositional ability. This gives a concrete example for the analysis in Theorem 1.

# E  Deferred Proof

In this section, we provide a formal setting and proof. We first formalize our model setup.

## E.1  Linear self-attention networks.

These networks are widely studied (Von Oswald et al., 2023; Akyürek et al., 2023; Mahankali et al., 2023; Garg et al., 2022; Zhang et al., 2023b; Shi et al., 2023b). Following them, we consider the following linear self-attention network with parameters $\theta = (W^{PV}, W^{KQ})$:

$$f_{\text{LSA},\theta}(E) = E + W^{PV} E \cdot \frac{E^\top W^{KQ} E}{N}.$$

The prediction of the model for $x_q$ is $\hat{y}_q = [f_{\text{LSA},\theta}(E)]_{(d+1):(d+K),N+1}$, the bottom rightmost sub-vector of $f_{\text{LSA},\theta}(E)$ with length $K$. Let

$$W^{PV} = \begin{pmatrix} W_{11}^{PV} & W_{12}^{PV} \\ (W_{21}^{PV})^\top & W_{22}^{PV} \end{pmatrix} \in \mathbb{R}^{(d+K)\times(d+K)}, W^{KQ} = \begin{pmatrix} W_{11}^{KQ} & W_{12}^{KQ} \\ (W_{21}^{KQ})^\top & W_{22}^{KQ} \end{pmatrix} \in \mathbb{R}^{(d+K)\times(d+K)},$$

where $W_{11}^{PV} \in \mathbb{R}^{d\times d}$, $W_{12}^{PV}, W_{21}^{PV} \in \mathbb{R}^{d\times K}$, and $W_{22}^{PV} \in \mathbb{R}^{K\times K}$; similar for $W^{KQ}$. Then the prediction is

$$\hat{y}_q = \left( (W_{21}^{PV})^\top \quad W_{22}^{PV} \right) \left( \frac{EE^\top}{N} \right) \begin{pmatrix} W_{11}^{KQ} \\ (W_{21}^{KQ})^\top \end{pmatrix} x_q. \tag{1}$$

We observe only part of the parameters affect our prediction, so we treat the rest of them as zero in our analysis.

## E.2  Proof of Compositional Ability under Confined Support

Here, we provide the proof of our main conclusion regarding Theorem 1 and Corollary 1.

Without abuse of notation, we denote $U = W_{11}^{KQ}, u = W_{22}^{PV}$.

We also add some mild assumptions.

1. The covariance matrix $\Lambda$ of simple tasks will have the same trace to prevent the scale effect of different simple tasks.

2. The spectral norm of $\Lambda$ is bounded on both sides $m \leq \|\Lambda\| \leq M$.

We first introduce the lemma where the language model only pretrained on one simple task ($K = 1$). The pretraining loss $L(\theta)$ can be refactored, and the solution will have a closed form. We further discuss the following.

**Lemma E.1** (Lemma 5.3 in Zhang et al. (2023b)). *Let* $\Gamma := \left(1 + \frac{1}{N}\right) \Lambda + \frac{1}{N} \operatorname{tr}(\Lambda) I_{d \times d} \in \mathbb{R}^{d \times d}$.
*Let*

$$\tilde{\ell}(U, u) = \operatorname{tr}\left[\frac{1}{2} u^2 \Gamma \Lambda U \Lambda U^\top - u \Lambda^2 U^\top\right]$$

*Then*

$$\min_\theta L(\theta) = \min_{U,u} \tilde{\ell}(U, u) + C = -\frac{1}{2} \operatorname{tr}[\Lambda^2 \Gamma^{-1}] + C$$

*where C is a constant independent with $\theta$. For any global minimum of $\tilde{\ell}$, we have $uU = \Gamma^{-1}$.*

As the above lemma construction, we denote the optimal solution as $W^{*PV}$ and $W^{*KQ}$. Taking one solution as $U = \Gamma^{-1}, u = 1$, we observe the minimizer of global training loss is of the form:

$$W^{*PV} = \begin{pmatrix} 0_{d \times d} & 0_d \\ 0_d^\top & 1 \end{pmatrix}, W^{*KQ} = \begin{pmatrix} \Gamma^{-1} & 0_d \\ 0_d^\top & 0 \end{pmatrix}. \tag{2}$$

We then prove our main theory Theorem 1 in Section 4.2, we first re-state below:

**Theorem 1.** *Consider distinct tasks $k$ and $g$ with corresponding examples $\mathcal{S}_k, \mathcal{S}_g$. If two tasks have confined support, and Assumption 1 is true, then with high probability, the model has the compositional ability as defined in Definition 1. Moreover,*

$$Acc_\theta(\mathcal{S}_k) + Acc_\theta(\mathcal{S}_g) \leq Acc_\theta(\mathcal{S}_{k \cup g}).$$

*Proof of Theorem 1.* WLOG, consider two simple tasks, $K = 2$. We have $x = (a, b)$, where $a \in \mathbb{R}^{d_1}, b \in \mathbb{R}^{d_2}, d_1 + d_2 = d$. Since $x$ only has large values on certain dimensions, it's equivalent to just consider corresponding dimensions in $w$, i.e., for simple task 1, we have $w^{(1)} = (w_a, w_{\delta b})$, for simple task 2, we have $w^{(2)} = (w_{\delta a}, w_b)$.

We have $x \sim \Lambda$, where:

$$\Lambda = \begin{pmatrix} \Lambda_{\mathbb{KK}} & \Lambda_{\mathbb{KG}} \\ \Lambda_{\mathbb{GK}} & \Lambda_{\mathbb{GG}} \end{pmatrix}$$

- Task 1: $x = (a, 0_{d_2})^\top + (0, b_\delta)^\top, y = (w_a^\top a, 0) + (0, w_{\delta b}^\top b_\delta)$.

- Task 2: $x = (0_{d_1}, b)^\top + (a_\delta, 0_{d_2})^\top, y = (0, w_b^\top b) + (w_{\delta a}^\top a_\delta), 0)$.

- Composed task: $x = (a, b)^\top + (a_\delta, b_\delta)^\top, y = (w_a^\top a, w_b^\top b) + (w_{\delta a}^\top a_\delta, w_{\delta b}^\top b_\delta)$.

The form of $E$ is,

$$E := \begin{pmatrix} a_1 & a_2 & \dots & a_N & a_q \\ b_1 & b_2 & \dots & b_N & b_q \\ y_1 & y_2 & \dots & y_N & 0 \end{pmatrix} + E_r \in \mathbb{R}^{(d+2) \times (N+1)}.$$

where $E_r$ represents the values caused by residual dimensions whose entries are bounded by $\delta$.

Following Equation (4.3) in Zhang et al. (2023b), we have

$$EE^\top = \frac{1}{N} \begin{pmatrix} \sum_{i=1}^N a_i a_i^\top + a_q a_q^\top & \sum_{i=1}^N a_i b_i^\top + a_q b_q^\top & \sum_{i=1}^N a_i y_i^\top \\ \sum_{i=1}^N b_i a_i^\top + b_q a_q^\top & \sum_{i=1}^N b_i b_i^\top + b_q b_q^\top & \sum_{i=1}^N b_i y_i^\top \\ \sum_{i=1}^N y_i a_i^\top & \sum_{i=1}^N y_i b_i^\top & \sum_{i=1}^N y_i y_i^\top \end{pmatrix} + \delta \cdot o(EE^\top).$$

The $W^{PV}$ can be presented in block matrix

$$W^{PV} = \begin{pmatrix} W_{11}^{PV} & W_{12}^{PV} & W_{13}^{PV} \\ (W_{21}^{PV})^\top & W_{22}^{PV} & W_{23}^{PV} \\ (W_{31}^{PV})^\top & (W_{32}^{PV})^\top & W_{33}^{PV} \end{pmatrix} \in \mathbb{R}^{(d_1+d_2+2)\times(d_1+d_2+2)}$$

We can apply Lemma E.1 into optimization and recall

$$W^{*KQ} = \begin{pmatrix} \Gamma_{all}^{-1} & 0_d \\ 0_d^\top & 0 \end{pmatrix}.$$

where $\Gamma_{all}^{-1} \in \mathbb{R}^{(d_1+d_2)\times(d_1+d_2)}$. Consider two tasks only related to disjoint dimension of $x$, we also have $\sigma(\Lambda_{\mathbb{K}\mathbb{G}}) = \sigma(\Lambda_{\mathbb{G}\mathbb{K}}) \leq \epsilon$. Denote

$$\Lambda = \tilde{\Lambda} + \Lambda_r$$

where

$$\tilde{\Lambda} = \begin{pmatrix} \Lambda_{\mathbb{K}\mathbb{K}} & \\ & \Lambda_{\mathbb{G}\mathbb{G}} \end{pmatrix}, \Lambda_r = \begin{pmatrix} & \Lambda_{\mathbb{K}\mathbb{G}} \\ \Lambda_{\mathbb{G}\mathbb{K}} & \end{pmatrix}$$

We apply Lemma E.1 Recall $\Gamma := \left(1 + \frac{1}{N}\right)\Lambda + \frac{1}{N}\operatorname{tr}(\Lambda)I_{d\times d} \in \mathbb{R}^{d\times d}$, we have:

$$\begin{aligned} \Gamma &= \left(1 + \frac{1}{N}\right)\tilde{\Lambda} + \frac{1}{N}\operatorname{tr}(\tilde{\Lambda})I_{d\times d} + \left(1 + \frac{1}{N}\right)\Lambda_r \\ &= \tilde{\Gamma} + \Gamma_r \end{aligned}$$

where denote $\Gamma_r = \left(1 + \frac{1}{N}\right)\Lambda_r$. We have:

$$\Gamma^{-1} = \tilde{\Gamma}^{-1} - \tilde{\Gamma}^{-1}\Gamma_r\tilde{\Gamma}^{-1} + \mathcal{O}(\Gamma_r)$$

We denote

$$\tilde{\Gamma} = \begin{pmatrix} \Gamma_1 & 0 \\ 0 & \Gamma_2 \end{pmatrix},$$

where $\Gamma_1 = \left(1 + \frac{1}{N}\right)\Lambda_{\mathbb{K}\mathbb{K}} + \frac{1}{N}\operatorname{tr}(\Lambda)I_{d_1} \in \mathbb{R}^{d_1\times d_1}$ and $\Gamma_2 = \left(1 + \frac{1}{N}\right)\Lambda_{\mathbb{G}\mathbb{G}} + \frac{1}{N}\operatorname{tr}(\Lambda)I_{d_2} \in \mathbb{R}^{d_2\times d_2}$. Then we have;

$$\Gamma^{-1} = \begin{pmatrix} \Gamma_1^{-1} & 0 \\ 0 & \Gamma_2^{-1} \end{pmatrix} + A$$

where $\sigma(A) \leq 2m^2\epsilon$.

Then, It's similar to applying Lemma E.1 for pretraining separately into dimensions corresponding to different tasks. We solve similarly to $W_{KQ}$.

We have:

$$f_\theta(E) = \begin{pmatrix} 0_{d_1\times d_1} & 0_{d_1\times d_2} & 0_{d_1\times 2} \\ 0_{d_2\times d_1} & 0_{d_2\times d_2} & 0_{d_2\times 2} \\ 0_{2\times d_1} & 0_{2\times d_2} & I_2 \end{pmatrix} EE^\top \begin{pmatrix} \Gamma_1^{-1} & 0_{d_1\times d_2} & 0_{d_1\times 2} \\ 0_{d_2\times d_1} & \Gamma_2^{-1} & 0_{d_2\times 2} \\ 0_{2\times d_1} & 0_{2\times d_2} & 0_{2\times 2} \end{pmatrix} \begin{pmatrix} a_q \\ b_q \\ 0 \end{pmatrix} + \tilde{A} \tag{3}$$

$$\hat{y}_q = \frac{1}{N}\left(\sum_{i=1}^N y_i a_i^\top, \sum_{i=1}^N y_i b_i^\top, \sum_{i=1}^N y_i y_i^\top\right)\begin{pmatrix} \Gamma_1^{-1}a_q \\ \Gamma_2^{-1}b_q \\ 0 \end{pmatrix} + v \tag{4}$$

$$= \left(\frac{1}{N}\sum_{i=1}^N y_i a_i^\top\right)\Gamma_1^{-1}a_q + \left(\frac{1}{N}\sum_{i=1}^N y_i b_i^\top\right)\Gamma_2^{-1}b_q + v \tag{5}$$

$$= \frac{1}{N}\begin{pmatrix} a_q^\top \Gamma_1^{-1}\sum_{i=1}^N y_i^{(1)}a_i \\ b_q^\top \Gamma_2^{-1}\sum_{i=1}^N y_i^{(2)}b_i \end{pmatrix} + v. \tag{6}$$

where $\tilde{A}$ representing residual matrix whose norm can be bounded by $\mathcal{O}(m^2\epsilon\delta)$ Recall $x \sim N(0, \Lambda)$, then with high probability each entry in $v$ will be bounded by $Cm^2\delta\epsilon$ for some constant $C$.

WLOG, we write residual vectors as 0 vector for simplicity of notation, and only consider residuals for estimations $\hat{y}$. Note that we composed example $x = (a, b)^\top$, $y = (w_a^\top a, w_b^\top b)$. For simplicity, we write $\hat{w}_a = \frac{1}{N}\Gamma_1^{-1}\sum_{i=1}^N y_i^{(1)} a_i$, similarly, $\hat{w}_b = \frac{1}{N}\Gamma_2^{-1}\sum_{i=1}^N y_i^{(2)} b_i$.

Given in-context examples from one simple task only, consider that we have $N$ examples from simple task 1, $\mathcal{S}_1 = \left[\{(a_i, 0), y_i\}_{i=1}^N\right]$. We have $\hat{w}^{(1)} = (\hat{w}_a, 0_{d_2})$, $\hat{w}^{(2)} = (0_d)$, and we also have $\hat{y}_q = (\hat{y}_q^{(1)}, 0)^\top$, where $\hat{y}_q^{(1)} = a_q^\top \Gamma_1^{-1}\left(\frac{1}{N}\sum_{i=1}^N y_i^{(1)} a_i\right) + Cm^2\delta\epsilon$. We have $\text{Acc}_\theta(\mathcal{S}_1) = \frac{\mathbb{1}\left(\tilde{y}_q^{(1)}=y_q^{(1)}\right)}{2}$.

Similarly, for $N$ in-context examples only from task 2, we have $\hat{w}^{(1)} = (0_d)$, $\hat{w}^{(2)} = (0_{d_1}, \hat{w}_b)$, $\hat{y}_q = (0, \hat{y}_q^{(2)})^\top$, where $\hat{y}_q^{(2)} = a_q^\top \Gamma_2^{-1}\left(\frac{1}{N}\sum_{i=1}^N y_i^{(2)} b_i\right) + Cm^2\delta\epsilon$. We have $\text{Acc}_\theta(\mathcal{S}_2) = \frac{\mathbb{1}\left(\tilde{y}_q^{(2)}=y_q^{(2)}\right)}{2}$.

Then we have $\mathcal{S}_{1\cup2}$ contains $2N$ in-context examples from both tasks, specifically, we have $N$ from task 1 and rest from task 2. We have $\hat{w}^{(1)} = (\hat{w}_a/2, 0_{d_2})$, $\hat{w}^{(2)} = (0_{d_1}, \hat{w}_b/2)$, $\hat{y}_q = (\hat{y}_q^{(1)}, \hat{y}_q^{(2)})^\top$.

Since $y_{\tau,q}^{(k)} = \text{sgn}(\langle w_\tau, x_{\tau,q}\rangle)$, $\tilde{y}_{\tau,q}^{(k)} = \text{sgn}\left(\hat{y}_{\tau,q}^{(k)}\right)$, following the proof of Lemma E.2, where $\text{Acc}_\theta$ only concerns the direction of $\hat{w}$ and $w$, we have $\text{Acc}_\theta(\mathcal{S}_{1\cup2}) = \frac{\mathbb{1}\left(\tilde{y}_q^{(1)}=y_q^{(1)}\right)+\mathbb{1}\left(\tilde{y}_q^{(2)}=y_q^{(2)}\right)}{2}$.

Extending the above analysis into any of two simple tasks, when the composite task integrates them, we have

$$\text{Acc}_\theta(\mathcal{S}_k) + \text{Acc}_\theta(\mathcal{S}_g) \leq \text{Acc}_\theta(\mathcal{S}_{k\cup g}). \tag{7}$$

$\square$

We then prove Corollary 1, and we first restate it below.

**Corollary 1.** *If two tasks do not have confined support, there exists one setting in which we have*

$$Acc_\theta(\mathcal{S}_k) = Acc_\theta(\mathcal{S}_g) = Acc_\theta(\mathcal{S}_{k\cup g}).$$

*Proof of Corollary 1.* WLOG, consider two simple tasks, $K = 2$. We have $x = (a, b)$, where $a \in \mathbb{R}^{d_1}, b \in \mathbb{R}^{d_2}, d_1 + d_2 = d$. Consider the setting where $w$ also have the same active dimensions, i.e., for simple task 1, we have $w^{(1)} = (w_a, 0)$, for simple task 2, we have $w^{(2)} = (0, w_b)$.

We have $x \sim \Lambda$. Consider tasks are overlapping on all dimensions, where:

- Task 1: $x = (a^{(1)}, b^{(1)})^\top$, $y = (w_a^\top a^{(1)}, w_b^\top b^{(1)})$.

- Task 2: $x = (a^{(2)}, b^{(2)})^\top$, $y = (w_a^\top a^{(2)}, w_b^\top b^{(2)})$.

- Composed task: $x = (a, b)^\top$, $y = (w_a^\top a, w_b^\top b)$.

Similarly, we have:

$$\hat{y}_q = \frac{1}{N} \left( \sum_{i=1}^{N} y_i a_i^\top, \sum_{i=1}^{N} y_i b_i^\top, \sum_{i=1}^{N} y_i y_i^\top \right) \begin{pmatrix} \Gamma_1^{-1} a_q \\ \Gamma_2^{-1} b_q \\ 0 \end{pmatrix} \tag{8}$$

$$= \left( \frac{1}{N} \sum_{i=1}^{N} y_i a_i^\top \right) \Gamma_1^{-1} a_q + \left( \frac{1}{N} \sum_{i=1}^{N} y_i b_i^\top \right) \Gamma_2^{-1} b_q \tag{9}$$

$$= \frac{1}{N} \begin{pmatrix} a_q^\top \Gamma_1^{-1} \sum_{i=1}^{N} y_i^{(1)} a_i + b_q^\top \Gamma_2^{-1} \sum_{i=1}^{N} y_i^{(1)} b_i \\ a_q^\top \Gamma_1^{-1} \sum_{i=1}^{N} y_i^{(2)} a_i + b_q^\top \Gamma_2^{-1} \sum_{i=1}^{N} y_i^{(2)} b_i \end{pmatrix}. \tag{10}$$

Note that composed example $x = (a, b)^\top$, $y = (w_1^\top a, w_2^\top b)$.

When in-context examples are from a simple task, we have $N$ examples from simple task 1, $\mathcal{S}_1 = \left[ \left\{ (a_i^{(1)}, b_i^{(1)}), y_i \right\}_{i=1}^{N} \right]$, and $\hat{y}_q$ has the same form as Equation (10), similarly, for task 2.

Suppose $\mathcal{S}_{1 \cup 2}$ contains $2N$ examples from both tasks, where $N$ from task 1 and rest from task 2. We have

$$\hat{y}_q = \frac{1}{2N} \begin{pmatrix} a_q^\top \Gamma_1^{-1} \sum_{i=1}^{N} y_i^{(1)} a_i + b_q^\top \Gamma_2^{-1} \sum_{i=1}^{N} y_i^{(1)} b_i \\ a_q^\top \Gamma_1^{-1} \sum_{i=1}^{N} y_i^{(2)} a_i + b_q^\top \Gamma_2^{-1} \sum_{i=1}^{N} y_i^{(2)} b_i \end{pmatrix}. \tag{11}$$

We finish the proof by checking that Equation (10) and Equation (11) share the same direction.

$\square$

### E.3 Proof of Compositional Ability with Model Scale

Here, we provide the proof of our conclusions in Theorem 2 in Appendix D.1 with respect to model performance and model scale. We first introduce a lemma under the $K = 1$ setting.

#### E.3.1 Accuracy under $K = 1$

When $K = 1$, we can give an upper bound of accuracy by $\Lambda$ and $\Gamma$. Taking into account the optimal solution in Equation (2), we have the following accuracy lemma.

**Lemma E.2.** *Consider $K = 1$ and $x_q \sim \mathcal{N}(0, I_d)$. When $N > C$, where $C$ is a constant, we have*

$$\mathbb{E}_{w_\tau, x_1, \cdots, x_N} [Acc_\theta] \leq \text{tr}(\Gamma^{-1} \Lambda).$$

*Proof of Lemma E.2.* Since $K = 1$, the problem reduces to the linear regression problem in ICL. Consider the solution form in Lemma E.1, we have

$$\hat{y}_q = x_q^\top \frac{1}{N} \Gamma^{-1} \sum_{i=1}^{N} \langle w_\tau, x_i \rangle x_i$$

We re-write the form as $\hat{y}_q = x_q^\top \hat{w}$. Following Equation (4.3) in Zhang et al. (2023b), we have:

$$\hat{w} = \frac{1}{N} \Gamma^{-1} \sum_{i=1}^{N} \langle w_\tau, x_i \rangle x_i.$$

Recall the definition of $Acc_\theta$ and $y_{\tau,q}^{(k)} = \text{sgn}(\langle w_\tau, x_{\tau,q} \rangle)$, $\tilde{y}_{\tau,q}^{(k)} = \text{sgn}\left( \hat{y}_{\tau,q}^{(k)} \right) = \text{sgn}(\langle \hat{w}, x_{\tau,q} \rangle)$, for any $\alpha > 0$, we have:

$$\mathbb{E}_{w_\tau, x_1, \cdots, x_N, x_q} [Acc_\theta] = P \left( \langle x_q, w_\tau \rangle > 0, \langle x_q, \alpha \hat{w} \rangle > 0 \right) + P \left( \langle x_q, w_\tau \rangle < 0, \langle x_i, \alpha \hat{w} \rangle < 0 \right).$$

Denote hyperplane orthogonal to $w$ as $\mathcal{P}_w$ and similar to $\mathcal{P}_{\hat{w}}$. Recall that $x_q$ is independent of other samples. We have the expectation conditioned on $w_\tau, x_1, \cdots, x_N$ is the probability that $x_q$ falls out of the angle between $\mathcal{P}_w$ and $\mathcal{P}_{\hat{w}}$. Denote the angle between $w$ and $\hat{w}$ as $\widetilde{\theta}$. As $x_q$ is uniform along each direction (uniform distribution or isotropic Gaussian), then the probability is $1 - \frac{|\widetilde{\theta}|}{\pi}$ given $w_\tau, x_1, \cdots, x_N$. Then $\mathbb{E}_{w_\tau, x_1, \cdots, x_N}[\text{Acc}_\theta] = \mathbb{E}_{w_\tau, x_1, \cdots, x_N}\left[1 - \frac{|\widetilde{\theta}|}{\pi}\right]$. Note that

$$\mathbb{E}_{w_\tau, x_1, \cdots, x_N}\left[\cos(\widetilde{\theta})\right] = \left\langle \frac{w_\tau}{\|w_\tau\|_2}, \frac{\hat{w}}{\|\hat{w}\|_2}\right\rangle.$$

As, we can choose $\alpha$, w.l.o.g, we take $\|w_\tau\| = \|\hat{w}\| = 1$, then we have

$$\mathbb{E}_{w_\tau, x_1, \cdots, x_N}\left[\cos(\widetilde{\theta})\right] = \mathbb{E}_{w_\tau}\left[\mathbb{E}_{x_1, \cdots, x_N}\left[\langle w_\tau, \hat{w}\rangle \,|w_\tau|\right]\right].$$

Given $w_\tau$, we have

$$\begin{aligned}
E[\hat{w}|w_\tau] &= \frac{1}{N}\Gamma^{-1}\sum_{i=1}^{N} E\left[\langle w_\tau, x_i\rangle x_i|w_\tau\right] \\
&= \frac{1}{N}\Gamma^{-1}\sum_{i=1}^{N}\Lambda w_\tau \\
&= \Gamma^{-1}\Lambda w_\tau.
\end{aligned}$$

Then, we have

$$\begin{aligned}
E_{w_\tau}\left[\langle \hat{w}, w_\tau\rangle\right] &= \left\langle \Gamma^{-1}\Lambda w_\tau^\top, w_\tau\right\rangle \\
&= \text{tr}(\Gamma^{-1}\Lambda).
\end{aligned}$$

Thus, we have

$$\mathbb{E}\cos(\widetilde{\theta}) = \text{tr}(\Gamma^{-1}\Lambda) \tag{12}$$

$$\mathbb{E}[\text{Acc}_\theta] = \mathbb{E}\left[1 - \frac{|\widetilde{\theta}|}{\pi}\right]. \tag{13}$$

Note that when $\theta \le \frac{\pi}{6}$, we have $1 - \frac{|\widetilde{\theta}|}{\pi} \le \cos(\theta)$. Thus, as $N > C$ where $C$ is constant, we have $\hat{w}$ and $w_\tau$ are closed and satisfy $\theta \le \frac{\pi}{6}$. Then we get the statement. $\qquad\square$

### E.3.2  Model scale on composite tasks

Here, we present proof for model scale and performance on composite tasks. Recall we consider the rank of $W^{*PV}$ and $W^{*KQ}$ as a measure of the model's scale.

We first introduce a lemma about $U$ as an optimal full-rank solution.

**Lemma E.3** (Corollary A.2 in Zhang et al. (2023b))**.** *The loss function $\widetilde{\ell}$ in Lemma E.1 satisfies*

$$\min_{U\in\mathbb{R}^{d\times d}, u\in\mathbb{R}} \widetilde{\ell}(U, u) = -\frac{1}{2}\text{tr}[\Lambda^2\Gamma^{-1}],$$

*where $U = c\Gamma^{-1}, u = \frac{1}{c}$ for any non-zero constant $c$ are minimum solution. We also have*

$$\widetilde{\ell}(U, u) - \min_{U\in\mathbb{R}^{d\times d}, u\in\mathbb{R}} \widetilde{\ell}(U, u) = \frac{1}{2}\left\|\Gamma^{\frac{1}{2}}\left(u\Lambda^{\frac{1}{2}}U\Lambda^{\frac{1}{2}} - \Lambda\Gamma^{-1}\right)\right\|_F^2. \tag{14}$$

As the scale of the model decreases, the rank of $U$ also reduces, leading to an optimal reduced rank solution $\widetilde{U}$. Our findings reveal that this reduced rank $\widetilde{U}$ can be viewed as a truncated form of the full-rank solution $U$. This implies that smaller-scale models are

essentially truncated versions of larger models, maintaining the core structure but with reduced complexity.

Recall $\Lambda$ is the covariance matrix, we have eigendecomposition $\Lambda = QDQ^\top$, where $Q$ is an orthonormal matrix containing eigenvectors of $\Lambda$ and $D$ is a sorted diagonal matrix with non-negative entries containing eigenvalues of $\Lambda$, denoting as $D = \mathrm{diag}([\lambda_1, \ldots, \lambda_d])$, where $\lambda_1 \geq \cdots \geq \lambda_d \geq 0$. We introduce the lemma below.

**Lemma E.4** (Optimal rank-$r$ solution). *Recall the loss function $\tilde{\ell}$ in (Lemma E.1). Let*

$$U^*, u^* = \underset{U \in \mathbb{R}^{d \times d}, \mathrm{rank}(U) \leq r, u \in \mathbb{R}}{\arg\min} \tilde{\ell}(U, u).$$

*Then $U^* = cQV^*Q^\top, u = \frac{1}{c}$, where $c$ is any non-zero constant and $V^* = \mathrm{diag}([v_1^*, \ldots, v_d^*])$ is satisfying for any $i \leq r, v_i^* = \frac{N}{(N+1)\lambda_i + \mathrm{tr}(D)}$ and for any $i > r, v_i^* = 0$.*

Then, we proof the Lemma E.4

*Proof of Lemma E.4.* Note that,

$$\underset{U \in \mathbb{R}^{d \times d}, \mathrm{rank}(U) \leq r, u \in \mathbb{R}}{\arg\min} \tilde{\ell}(U, u) = \underset{U \in \mathbb{R}^{d \times d}, \mathrm{rank}(U) \leq r, u \in \mathbb{R}}{\arg\min} \tilde{\ell}(U, u) - \underset{U \in \mathbb{R}^{d \times d}, u \in \mathbb{R}}{\min} \tilde{\ell}(U, u)$$

$$= \underset{U \in \mathbb{R}^{d \times d}, \mathrm{rank}(U) \leq r, u \in \mathbb{R}}{\arg\min} \left( \tilde{\ell}(U, u) - \underset{U \in \mathbb{R}^{d \times d}, u \in \mathbb{R}}{\min} \tilde{\ell}(U, u) \right).$$

Thus, we may consider Equation (14) in Lemma E.3 only. On the other hand, we have

$$\Gamma = \left(1 + \frac{1}{N}\right) \Lambda + \frac{1}{N} \mathrm{tr}(\Lambda) I_{d \times d}$$

$$= \left(1 + \frac{1}{N}\right) QDQ^\top + \frac{1}{N} \mathrm{tr}(D) QI_{d \times d} Q^\top$$

$$= Q \left( \left(1 + \frac{1}{N}\right) D + \frac{1}{N} \mathrm{tr}(D) I_{d \times d} \right) Q^\top.$$

We denote $D' = \left(1 + \frac{1}{N}\right) D + \frac{1}{N} \mathrm{tr}(D) I_{d \times d}$. We can see $\Lambda^{\frac{1}{2}} = QD^{\frac{1}{2}}Q^\top$, $\Gamma^{\frac{1}{2}} = QD'^{\frac{1}{2}}Q^\top$, and $\Gamma^{-1} = QD'^{-1}Q^\top$. We denote $V = uQ^\top UQ$. Since $\Gamma$ and $\Lambda$ are commutable and the Frobenius norm (F-norm) of a matrix does not change after multiplying it by an orthonormal matrix, we have Equation (14) as

$$\tilde{\ell}(U, u) - \underset{U \in \mathbb{R}^{d \times d}, u \in \mathbb{R}}{\min} \tilde{\ell}(U, u) = \frac{1}{2} \left\| \Gamma^{\frac{1}{2}} \left( u\Lambda^{\frac{1}{2}} U\Lambda^{\frac{1}{2}} - \Lambda\Gamma^{-1} \right) \right\|_F^2$$

$$= \frac{1}{2} \left\| \Gamma^{\frac{1}{2}} \Lambda^{\frac{1}{2}} \left( uU - \Gamma^{-1} \right) \Lambda^{\frac{1}{2}} \right\|_F^2$$

$$= \frac{1}{2} \left\| D'^{\frac{1}{2}} D^{\frac{1}{2}} \left( V - D'^{-1} \right) D^{\frac{1}{2}} \right\|_F^2.$$

As $W^{KQ}$ is a matrix whose rank is at most $r$, we have $V$ is also at most rank $r$. Then, we denote $V^* = \arg\min_{V \in \mathbb{R}^{d \times d}, \mathrm{rank}(V) \leq r} \left\| D'^{\frac{1}{2}} D^{\frac{1}{2}} \left( V - D'^{-1} \right) D^{\frac{1}{2}} \right\|_F^2$. We can see that $V^*$ is a diagonal matrix. Denote $D' = \mathrm{diag}([\lambda_1', \ldots, \lambda_d'])$ and $V^* = \mathrm{diag}([v_1^*, \ldots, v_d^*])$. Then, we have

$$\left\| D'^{\frac{1}{2}} D^{\frac{1}{2}} \left( V - D'^{-1} \right) D^{\frac{1}{2}} \right\|_F^2 \tag{15}$$

$$= \sum_{i=1}^{d} \left( \lambda_i'^{\frac{1}{2}} \lambda_i \left( v_i^* - \frac{1}{\lambda_i'} \right) \right)^2 \tag{16}$$

$$= \sum_{i=1}^{d} \left( \left( 1 + \frac{1}{N} \right) \lambda_i + \frac{\mathrm{tr}(D)}{N} \right) \lambda_i^2 \left( v_i^* - \frac{1}{\left( 1 + \frac{1}{N} \right) \lambda_i + \frac{\mathrm{tr}(D)}{N}} \right)^2. \tag{17}$$

As $V^*$ is the minimum rank $r$ solution, we have that $v_i^* \geq 0$ for any $i \in [d]$ and if $v_i^* > 0$, we have $v_i^* = \frac{1}{\left(1+\frac{1}{N}\right)\lambda_i + \frac{\text{tr}(D)}{N}}$. Denote $g(x) = \left(\left(1+\frac{1}{N}\right)x + \frac{\text{tr}(D)}{N}\right)x^2\left(\frac{1}{\left(1+\frac{1}{N}\right)x + \frac{\text{tr}(D)}{N}}\right)^2 = x^2\left(\frac{1}{\left(1+\frac{1}{N}\right)x + \frac{\text{tr}(D)}{N}}\right)$. It is easy to see that $g(x)$ is an increasing function on $[0, \infty)$. Now, we use contradiction to show that $V^*$ only has non-zero entries in the first $r$ diagonal entries. Suppose $i > r$, such that $v_i^* > 0$, then we must have $j \leq r$ such that $v_j^* = 0$ as $V^*$ is a rank $r$ solution. We find that if we set $v_i^* = 0, v_j^* = \frac{1}{\left(1+\frac{1}{N}\right)\lambda_j + \frac{\text{tr}(D)}{N}}$ and all other values remain the same, Equation (17) will strictly decrease as $g(x)$ is an increasing function on $[0, \infty)$. Thus, here is a contradiction. We finish the proof by $V^* = uQ^\top U^* Q$. $\qquad\square$

We then ready to prove the Theorem 2 in Appendix D.1, we first re-state it below.

**Theorem 2.** *Suppose a composite task satisfies confined support. Suppose that we have* $(x_1, y_1, \ldots, x_N, y_N, x_q)$ *as a testing input prompt and the corresponding $W$ where $y_i = Wx_i$. As rank $r$ decreases, $\mathbb{E}_{W, x_1, \cdots, x_N}[Acc_\theta]$ will have a smaller upper bound.*

*Proof of Theorem 2.* We first prove in a simple task setting ($K = 1$), that the accuracy will have such a conclusion. By Lemma E.2, consider $x_q \sim \mathcal{N}(0, I_d)$. When $N > C$, where $C$ is a constant, we have

$$\mathbb{E}_{w_\tau, x_1, \cdots, x_N}[Acc_\theta] \leq \text{tr}(\Gamma^{-1}\Lambda).$$

Recall Lemma E.4. WLOG, we take $c = 1$. We have

$$\text{tr}(\Gamma^{-1}\Lambda) = \text{tr}\left(QV^*DQ\right)$$

$$= \sum_{i=1}^{r} \frac{N}{N + 1 + \sum_{j=1}^{r}\frac{\lambda_j}{\lambda_i}},$$

where second equation comes from Lemma E.4.

Under the confined support setting, the same conclusion holds since Equation (7) in the proof of Theorem 1.

$\qquad\square$

