# OpenReview forum: "Do Large Language Models Have Compositional Ability? An Investigation into Limitations and Scalability"
_colmweb.org/COLM/2024/Conference — COLM_

### Official Review · Reviewer_EV4J · 2024-05-06

**Rating:** 6
**Confidence:** 4
**Ethics Flag:** 1

**Summary:**

This paper is targeting wheter LLMs have compositional ability through in-context learning setting. The paper get intuition from the observation of whether LLMs can solve comsite task formed by simple tasks and make two important contributions to this research question (1) empirically, they introduce several composite tasks to explore how the nature of the seperate task influence compisitonal performance; (2) theoretically, they introduce a simple yet insightful model to provide analysis. Based on these effort, the authors hightlight the variable performance of LLMs on these tasks and theoretical insights into conditions under which LLMs manage to achieve or fail in compositional ability.

**Reasons To Accept:**

1.This research question, compositional ability, is an important topic in the domain of LLMs and is critical for advancing towards more general AI capabilities.
2.The authors provide clear and insightful empirical experiments of when LLMs succeed or fail on compositional ability across a variety of tasks.
3.The authors have effectively linked the concept of compositional ability with scaling laws, demonstrating that when a model exhibits confined support for simple tasks, enhancements in model scale can yield superior performance. This finding is particularly useful for practical applications.

**Reasons To Reject:**

1.The theoretical section lacks a bridge that connects it effectively with the empirical results presented earlier. For instance, the first sentence, 'Despite the complex nature of non-linearity in transformers in LLMs,' does not seem to directly follow from or reference the empirical findings discussed previously
2.Given the extensive and diverse nature of pretraining data (most of them are closed source), there exists a possibility that certain components of the testing tasks may have been indirectly encountered by the models during pretraining phase. So I wonder how do authors ensure this in Section 3: "We make sure composite tasks have not been seen by the LLMs during pretraining.". This should be explained
3.The theoretical conclusions presented in the paper are derived based on certain assumptions regarding the models and data. However, the actual scenarios involving LLMs and their pretraining processes are notably complex. So, my question is whether there is any experimental evidence to validate the application of the proposed theorem for guiding real-world LLMs in tackling combinatorial tasks.

---

> ### Author Rebuttal · Authors · 2024-05-29
>
> We thank the reviewer for the valuable comments and address the questions below.
>
> > The theoretical section lacks an explanation on the non-linearity in transformers.
> We acknowledge that our linear self-attention framework is a simplification of actual transformers, but it captures core theoretical insights while remaining practical. We consider each input mapped to a feature vector. For simpler composite tasks that apply mapping to different input segments, the features of the inputs can be divided into disjoint regions. Conversely, the input features become more mixed and inseparable for more complex tasks involving sequential reasoning.
>
> This simplified framework aligns with recent works [1-4] analyzing in-context learning (ICL) on transformers. It provides a straightforward theoretical construction without compromising the fundamental nature of LLMs.
>
> > Explain ``composite tasks have not been seen by the LLMs during pretraining.”
>
> In this work, we opt for synthetic data due to the avoidance of data leakage. While we cannot fully ensure that the test data has not been seen during pretraining, we mitigate this risk by constructing our compositional test data using a unique syntax and mapping mechanism. This approach significantly shifts the data distribution away from existing web-scale data, making it highly improbable that our test data has been encountered during pretraining.
>
> We hope this explanation addresses your concern .
>
>
> > Experimental evidence to validate the application of the proposed theorem.
> We appreciate the reviewer for highlighting the connection between our theoretical conclusions and the real-world experimental results.
>
> We have provided an intuitive demonstration showing that inputs from simple tasks can be divided into distinct segments (e.g., (A) + (F)). This corresponds to the separability of the input features learned by the models. In future work, we plan to include visualizations of the learned embeddings of input, illustrating the feature space corresponding to the simple task inputs.
>
> [1] ​​Akyürek, Ekin et al. What learning algorithm is in-context learning? Investigations with linear models.
>
> [2] Oswald, Johannes Von et al. Transformers Learn In-Context by Gradient Descent.
>
> [3] Zhang, Ruiqi et al. Trained Transformers Learn Linear Models In-Context
>
> [4] Mahankali, Arvind et al. One Step of Gradient Descent is Provably the Optimal In-Context Learner with One Layer of Linear Self-Attention.

---

### Official Review · Reviewer_sUH4 · 2024-05-11

**Rating:** 6
**Confidence:** 4
**Ethics Flag:** 1

**Summary:**

This paper studies the compositional abilities in language models through in-context learning. In this study, the author introduces simple tasks who are quite intuitive to humans to analyze the reasoning abilities of LMs. The authors test these tasks on several LMs like GPT, Llama and Mistral with varying sizes. The results show that (i) models show some compositionality skills which improved with model sizes. (ii) models struggle on tasks requiring sequential reasoning. In addition to empirical analysis the authors also provide theoretical analysis.

**Questions To Authors:**

1. Given that the tasks are not present in the pretraining data, there is a performance improvement for increased sizes. Do these results correlate to better learning abilities of LMs?
2.  Do the authors think these results follow similar trend for all smaller LMs eg: BERT?
3. The authors claim that the tasks are not present in the training data. How do they validate it?
4. The authors missed a couple of related studies in their papers. How does these studies vary from the current proposed one?

a. Madasu, Avinash, and Shashank Srivastava. "What do Large Language Models Learn beyond Language?." In Findings of the Association for Computational Linguistics: EMNLP 2022, pp. 6940-6953. 2022.

b. Alajrami, Ahmed, Katerina Margatina, and Nikolaos Aletras. "Understanding the Role of Input Token Characters in Language Models: How Does Information Loss Affect Performance?." In Proceedings of the 2023 Conference on Empirical Methods in Natural Language Processing, pp. 9085-9108. 2023.

**Reasons To Accept:**

1. The compositional tasks introduced are quite intuitive and can be used to understand compositional abilities of LMs.
2. The experiments are interesting and thorough. Authors also provided reasonable theoretical analysis for the behaviors.
3. The paper is clearly written and easy to follow. I enjoyed reading the paper.

**Reasons To Reject:**

None

---

> ### Author Rebuttal · Authors · 2024-05-29
>
> We thank the reviewer for the valuable comments and address the questions below.
> > There is a performance improvement for increased sizes. Do these results correlate to better learning abilities of LMs?
>
> Yes. Larger models indeed have better abilities. This has been shown in various related works [1,2]. Large-scale models will generally have better reasoning and learning abilities on various tasks.
>
>
> > Similar trend for all smaller LMs eg: BERT
>
> Yes, we conjecture smaller LMs like BERT will also have such a trend (better abilities for larger size). In our work we didn’t verify this since BERT is an encoder model and does not have zero-shot or in-context learning ability. The adaptation of models like BERT will require additional finetuning and is not covered by our work.
>
>
> > Other related works [a,b]
>
> We thank the reviewer for providing related studies [a,b], and we state the distinctions below.
>
> [a,b] primarily investigate the effects of pretraining on language models. Specifically, [a] examines whether pretraining enables language models to acquire reasoning skills. [b] focuses on how information loss in training token characters affects language model performance. In contrast, our work explores whether language models can learn reasoning through in-context learning without delving into pretraining aspects.
>
> Regarding the experimental models, [a, b] mainly experiment on relatively small models, including the encoder model (BERT, DeBERTa) and LSTM model (ELMO), while our work mainly explores large decoder-based language models, including Mistral, Llama, and open source GPT models (GPT-Neo, GPT-j-6b).
>
> We thank the reviewer's suggestions and will add these works to our related work in revision.
>
> > Validate the tasks are not present in the training data.
>
> In this work, we opt for synthetic data to avoid data leakage. While we cannot fully ensure that the test data has not been seen during pretraining, we mitigate this risk by constructing our compositional test data using a unique syntax and mapping mechanism. This approach significantly shifts the data distribution away from existing web-scale data, making it highly improbable that our test data has been encountered during pretraining.
>
> We hope this explanation helps address your concern.
>
>
> [1] Jerry Wei . et al. Larger language models do in-context learning differently.
>
> [2] Jason Wei . et al. Emergent abilities of large language models.

---

### Official Review · Reviewer_Biao · 2024-05-14

**Rating:** 6
**Confidence:** 3
**Ethics Flag:** 1

**Summary:**

In this paper, the authors investigate the compositional ability of LLMs. They found "(1) For simpler composite tasks that apply distinct mapping mechanisms to different input segments, the models demonstrate decent compositional ability while scaling up the model enhances this ability; (2) for more complex composite tasks that involve reasoning multiple steps, which each step represent one task, models typically underperform, and scaling up does not generally lead to improvements." They offer a theoretical analysis explaining the model when the task handles different input parts separately, the model exhibits compositional capability.

This paper provides an angle of how LLM does compositional on a toy task. The paper is written and it is novel. This work may benefit the understanding of LLM.

**Reasons To Accept:**

1. Interesting explanation of how LLM do compositional tasks.
2. Good theoretical analysis.

**Reasons To Reject:**

1. Need more evidence on more complex tasks for example mathematical reasonings.

---

> ### Author Rebuttal · Authors · 2024-05-29
>
> We thank the reviewer for the valuable comments and address the questions below.
> > Need more evidence on more complex tasks for example mathematical reasonings.
>
> We thank the reviewers for bringing up more tasks and models.
> In this work, we chose synthetic data to avoid the risk of data leakage. Due to time constraints, we only constructed the tasks presented in the paper. In future work, we plan to expand our task set, particularly focusing on mathematical reasoning. Additionally, we have included results from Llama3 for a more comprehensive evaluation.
> Due to time and budget limits, we added Llama3 results for part of our tasks to demonstrate the idea.
> |            | Tasks                | 8B | 70B |
> |------------|----------------------|----|-----|
> | (A) + (B)  | Capitalization       | 100| 100 |
> | -           | Swap                 | 100| 100 |
> | -           | Compose              |  52|  72 |
> | -           | Com. in-context      |  97| 100 |
> | (A) + (F)  | Capitalization      | 100| 100 |
> | -           | PlusOne  | 100| 100 |
> | -           | Compose              |  88|  100 |
> | -           | Com. in-context      |  100| 100 |
>
> As shown in the table, for the _separable composite tasks_ which are relatively easy for model to solve _(A) + (F)_, the models show strong compositional ability: the composite accuracy is high, improves with increasing scale, and eventually reaches similar performance as the _gold standard_ composite in-context setting.
>
> For composite tasks with sequential reasoning steps _(A) + (B)_,  the model has poor performance in small scale but have increased performance in increased model scale. Providing composed examples as in-context demonstrations will help the model understand the composite tasks and solve them well.

---

### Official Review · Reviewer_YRjW · 2024-05-20

**Rating:** 8
**Confidence:** 3
**Ethics Flag:** 1

**Summary:**

This paper studies the capabilities of LLMs in solving compositional reasoning tasks -- a crucial capability towards advancement of understanding capabilities of LLMs . They develop a test suite of composite tasks which differ in various linguistic and logical challenges.

In the composite logical challenge - they looked at a mix or word and numerical transformations. It was evaluated in 4 settings, the two simple tasks, the composite task with simple tasks as in-context examples and composite task with the composite tasks as in-context example. They find that separable composite tasks, where the simple tasks forming the composite task are distinct, are more straightforward to handle by LLMs, so compose-by-parts tasks seemed easier than compose-by-steps tasks. They also show that composite-in-context examples are easier to handle.

In the composite linguistic challenge - they evaluated the ability to translate english based on some customized rules. They constructed two tasks and was also evaluated in the same 4 settings. The authors find that LLMs are capable of handling these composite tasks, and their performance improves with increasing model scale. They attribute this success to the fact that these composite tasks are separable.

They support their results through theoretical analysis too where they say for separable composite tasks can occupy different subspaces for the individual simpler tasks in their input embeddings. They also discuss that larger models, with higher rank matrices, are better at compositional capabilities.

**Questions To Authors:**

1) I notice that the composite linguistic capabilities generalized much better than the logical ones, could some of the regressions also be attributed to some of the models numerical and code understanding cabilities?

2) When the in-context examples are of the actual composite task instead of the simpler tasks the generalization seems to be much better. I wonder if this implies that the way the in-context examples or the prompt instruction given to the model was tuned. Did the authors try some prompt tuning?

3) If some of the composite logical examples were made more natural language - I wonder if these would have changed things. Specially given my point (1). I'd be curious to know if you tried this?

**Reasons To Accept:**

This is a well written paper that attempts to understand a very challenging problem for LLMs, i.e. compositional generalization.

I like how the authors tried to isolate this task and approached it in a systematic approach. They provide support with both empirical and theoretical findings. Although I still have questions around if the tasks have actually been isolated, I believe the paper's findings could stimulate further research into LLM reasoning and compositional generalization.

**Reasons To Reject:**

The approach seems like something that could have been done on top of more commercial and bigger LLMs too, like GPT, LLama-3, Claude and Gemini. I would really like to know how these capabilities scale to those models. Specially they seem to be particularly better at numeric transformations and code-understanding.

---

> ### Author Rebuttal · Authors · 2024-05-29
>
> We thank the reviewer for the valuable comments and address the questions below.
> > Composite linguistic capabilities generalized better than the logical ones, could some regressions also be attributed to the models numerical and code understanding capabilities?
>
> Yes. We believe natural language inputs can indeed help language models understand concepts better than special symbols or code. Natural language provides a richer context, which aligns better with the way these models are trained on large text corpora.  In contrast, logical and numerical tasks often rely on more rigid structures, which is harder for models to generalize without explicit training on such patterns.
> >  Did the authors try some prompt tuning?
>
> We've experimented with prompt demonstrations. Instructional prompts do help ChatGPT and Claude3 (although we haven't quantified the accuracy in large-scale experiments), but they offer limited benefits for current open-source models. On the other hand, we did not have prompt tuning or any other parameter updates during our evaluation.
>
> In our experiments, we provided the model with instructions. Here are instructions of Figure 1, which were prepended to ICL examples.
> ```
> * is a function for capitalizing letters, () is another function for swapping the position of 2 words.
> ```
> We will add full instructions to the appendix in further revisions.
>
> > What if some of the composite logical examples were made more natural language.
>
> We believe it is an interesting direction. Rephrasing the tasks into natural language could leverage the model's strengths in understanding and processing human language, possibly improving its performance in logical reasoning. We will explore it in our future work.
>
> > More commercial and bigger LLMs.
>
> Due to time and budget limits, we added Llama3 results for part of our tasks to demonstrate the idea.
>
> |            | Tasks                | 8B | 70B |
> |------------|----------------------|----|-----|
> | (A) + (B)  | Capitalization       | 100| 100 |
> | -           | Swap                 | 100| 100 |
> | -           | Compose              |  52|  72 |
> | -           | Com. in-context      |  97| 100 |
> | (A) + (F)  | Capitalization      | 100| 100 |
> | -           | PlusOne  | 100| 100 |
> | -           | Compose              |  88|  100 |
> | -           | Com. in-context      |  100| 100 |
>
> LLMs perform better on ``separable composite tasks'' (A)+(F) but poorly on composite tasks with sequential reasoning steps (A) + (B).

---

> > ### Comment · Reviewer_YRjW · 2024-06-06
> >
> > Thank you for your comments. I will keep my score.

---

### Decision · Program_Chairs · 2024-07-10

**Decision:**

Accept

**Comment:**

This paper studies the ability of LLMs to perform compositional reasoning through in-context learning.

Scores are positive, one very positive and three lukewarm but still above acceptance threshold.

Strengths:
- well-motivated study of important problem (YRjW, Biao, EV4J)
- the problem is isolated and studied systematiicaly (YRjW)
- theoretical analysis (Biao, EV4J)
- well-motivated tasks (sUH4)
- thorough experiments (sUH4)

Weaknesses:
- no inclusion of large/commerical LLMs, beyond partial inclusion of LLaMa3 during rebuttal (YRjW)
- despite synthetic data, some reviewers voiced concerns about data leakage (e.g. EV4J), which the authors argue is addressed by using a custom syntax
- simplistic theoretical analysis compared to real transformers (EV4J), which the authors is in line with other recent work on ICL

One reviewer asked for more tasks (Biao), though with little specific indication of which these would be.